# Site-specific phosphorylation of TRANSPARENT TESTA GLABRA1 mediates carbon partitioning in *Arabidopsis* seeds

Chengxiang Li [1], Bin Zhang [1], Bin Chen [2], Lianghui Ji [2] & Hao Yu [1,2]

Seed development is dependent on nutrients, such as a source of carbon, supplied by the parent plant. It remains largely unknown how these nutrients are distributed to zygotic and maternal tissues to coordinate storage of reserve compounds and development of protective tissues like seed coat. Here we show that phosphorylation of TRANSPARENT TESTA GLABRA1 (TTG1) is regulated by SHAGGY-like kinases 11/12 (SK11/12) and that this mediates carbon flow to fatty acid synthesis and seed coat traits in *Arabidopsis* seeds. SK11/12 phosphorylate TTG1 at serine 215, thus preventing TTG1 interaction with TRANSPARENT TESTA2. This compromises recruitment of TTG1 to the *GLABRA2* locus and downregulates *GLABRA2* expression, which enhances biosynthesis of fatty acids in the embryo, but reduces production of mucilage and flavonoid pigments in the seed coat. Therefore, site-specific phosphorylation of TTG1 by SK11/SK12 regulates carbon partitioning between zygotic and maternal sinks in seeds.

---

[1] Department of Biological Sciences, Faculty of Science, National University of Singapore, Singapore, 117543, Singapore. [2] Temasek Life Sciences Laboratory, National University of Singapore, Singapore, 117604, Singapore. Correspondence and requests for materials should be addressed to H.Y. (email: dbsyuhao@nus.edu.sg)

Seed development is a key process during which seed plants accumulate storage reserves in zygotic tissues that not only support seed germination and post-germinative seedling growth, but also serve as essential dietary nutrients for human beings and animals. The zygotic embryo and endosperm are usually covered by maternally derived tissues, such as seed coat, which protect the seed from external stresses to facilitate the balance between seed dormancy and germination[1].

In the model angiosperm plant *Arabidopsis thaliana*, reserve compounds in mature seeds are mostly stored in the zygotic embryo and mainly comprise storage proteins and oil in the form of triacylglycerols consisting of esters of glycerol and fatty acids[1,2]. While accumulation of these seed storage reserves is dependent on availability of nutrients supplied by the parent plant, it is highly subject to competition with other metabolic pathways in seeds that share the same nutrients. Sucrose is produced by photosynthetic tissues in the parent plant, and serves as a major carbon source for accumulation of seed storage oil and generation of other important seed components, such as seed coat mucilage and pigments[1,3,4]. Hydrolysis of sucrose produces glucose, which is converted by glycolysis into acetyl-coenzyme A (CoA) that can be further carboxylated into malonyl-CoA. Both acetyl-CoA and malonyl-CoA are key substrates for fatty acid biosynthesis in the embryo[1], while malonyl-CoA also serves as a major substrate for biosynthesis of flavonoid pigments mainly consisting of proanthocyanidins (PAs) in the seed coat and flavonols in both the seed coat and the embryo in *Arabidopsis* seeds[5]. Meanwhile, glucose can be converted into rhamnose for further synthesis of rhamnogalaturonan I, which constitutes pectin, a primary component in seed coat mucilage[6–8]. Therefore, production of seed oil, pigments, and seed coat mucilage in zygotic and maternal tissues shares the same carbon source, indicating a regulatory demand for coordinating carbon partitioning among these metabolic pathways during seed development.

*TRANSPARENT TESTA GLABRA1* (*TTG1*) encodes a WD40 repeat transcription factor that plays multiple roles in affecting post-embryonic development and seed development in *Arabidopsis*[9–14]. It functions post-embryonically in determining the formation of hairs on leaves, stems, and roots. During seed development, TTG1 simultaneously play distinct roles in affecting several key traits that share the same carbon source. It promotes biosynthesis of flavonoids, including PAs that are mainly present in the seed coat. Consequently, *ttg1* loss-of-function mutants exhibit the transparent testa phenotype as a result of low levels of oxidized PAs in the seed coat[5,9–11]. TTG1 is also required for the production of mucilage and the associated columella, both of which are absent in the seed coat of *ttg1* mutants[9,15,16]. In contrast, TTG1 plays a negative role in mediating the accumulation of seed storage reserves including fatty acids during the seed maturation process[17]. These findings imply that TTG1 could act as a key regulator that mediates carbon partitioning among various metabolic fluxes in zygotic and maternal tissues during seed development.

Previous studies have indicated that TTG1 could interact with other MYB and basic helix-loop-helix (bHLH) transcription factors to form MYB-bHLH-WD40 ternary complexes that regulate the expression of other downstream genes involved in various developmental processes[18–21]. For example, it has been suggested that the complex consisting of TRANSPARENT TESTA2 (TT2; MYB123), TRANSPARENT TESTA8 (TT8; bHLH042), and TTG1 regulates the expression of *BANYULS* that encodes a core enzyme for PA biosynthesis in the seed coat[18]. In addition, a potential target of TTG1, *GLABRA2* (*GL2*)[21], also affects TTG1-relevant seed characters[3,22–24]. However, as experimental evidence for TTG1 interaction with other partners

or its binding to downstream targets in *Arabidopsis* seeds have so far not yet been obtained, the exact in planta functions of TTG1 in molecular control of seed development remain largely unknown.

Here we show that two glycogen synthase kinase-3 (GSK3)-like kinases, SHAGGY-like kinases 11/12 (SK11/SK12), phosphorylate TTG1 to enhance fatty acid biosynthesis in the embryo while inhibiting production of mucilage and flavonoid pigments in the seed coat. We propose that phosphorylation of TTG1 by SK11/SK12 represents a molecular framework to modulate carbon partitioning in zygotic and maternal seed tissues.

## Results

**SK11 and SK12 interact with TTG1.** The TTG1 protein contains four typical WD40 repeats[10], which were predicted to fold into a canonical seven-blade β-propeller[25] (Supplementary Fig. 1) that could play a role in mediating protein–protein interactions through multiple surfaces of the β-propeller as shown in common WD40 proteins[26]. To identify new interacting partners of TTG1 in seed development, we used the recombinant MBP-TTG1 protein to pull down its interacting proteins in total protein extracts from *Arabidopsis* Col-0 wild-type siliques. After analyzing the co-purified complexes by liquid chromatography coupled with tandem mass spectrometry (LC–MS/MS), we identified a glycogen synthase kinase-3 (GSK3)-like kinase, SHAGGY-like kinases11 (SK11; At5g26751), as a potential interacting protein of TTG1 (Supplementary Fig. 2). There are a total of 10 GSK3-like kinases that are divided into four subgroups in *Arabidopsis*, which may play diverse roles in plant developmental signaling pathways[27]. SK11 shares the highest sequence similarity with SK12 (At3g05840) in the subgroup I, implying that they may have overlapping biological functions.

We then performed a detailed analysis of the interaction between TTG1 and SK11 or SK12 using various approaches. Glutathione S-transferase (GST) pull-down assays demonstrated that GST-SK11 and GST-SK12 bound to in vitro-translated MBP-TTG1, but not MBP itself (Fig. 1a). Bimolecular fluorescence complementation (BiFC) assays revealed the yellow fluorescent protein (YFP) fluorescence signal in both the nucleus and cytoplasm of tobacco epidermal cells (Fig. 1b), implying a direct interaction between TTG1 and SK11 or SK12 in living plant cells. Subcellular localization of TTG1-mCherry in tobacco epidermal cells and immunoblot analysis of 4HA-TTG1 (Supplementary Fig. 3) in proteins extracted from *ttg1-13 g4HA-TTG1* (Supplementary Fig. 4) *Arabidopsis* siliques confirmed localization of TTG1 in both the nucleus and cytoplasm. Yeast two-hybrid assays also showed the interaction of SK11 and SK12 with TTG1 in yeast cells (Fig. 1c).

To verify the interaction between TTG1 and SK11 in *Arabidopsis* seeds, we created two types of transgenic lines, *ttg1-13 g4HA-TTG1*, and *SK11:SK11^{E292K}-GFP*. First, we transformed the Col-0 near-isogenic *ttg1-13* line, which was obtained by three backcrosses of the deletion mutant *ttg1-13* in the Rschew (RLD) background[10] into Col-0 wild-type plants, with a 4.6-kb *TTG1* genomic fragment including the 2.5-kb upstream sequence, the 1.0-kb coding sequence harboring a 4HA tag fused immediately after ATG, and the 1.1-kb downstream sequence. Most of *ttg1-13 g4HA-TTG1* transformants displayed the seed phenotypes like wild-type plants (Supplementary Fig. 4a, b), implying that the 4HA-TTG1 fusion protein is biologically functional. One representative line (*#1-1*) that could contain only one T-DNA insertion site based on the segregation ratio was selected for further investigation. Second, we created transgenic plants expressing *SK11-GFP* and *SK11^{E292K}-GFP* driven by a 2.6-kb *SK11* 5′ upstream sequence. SK11^{E292K}-GFP contained a

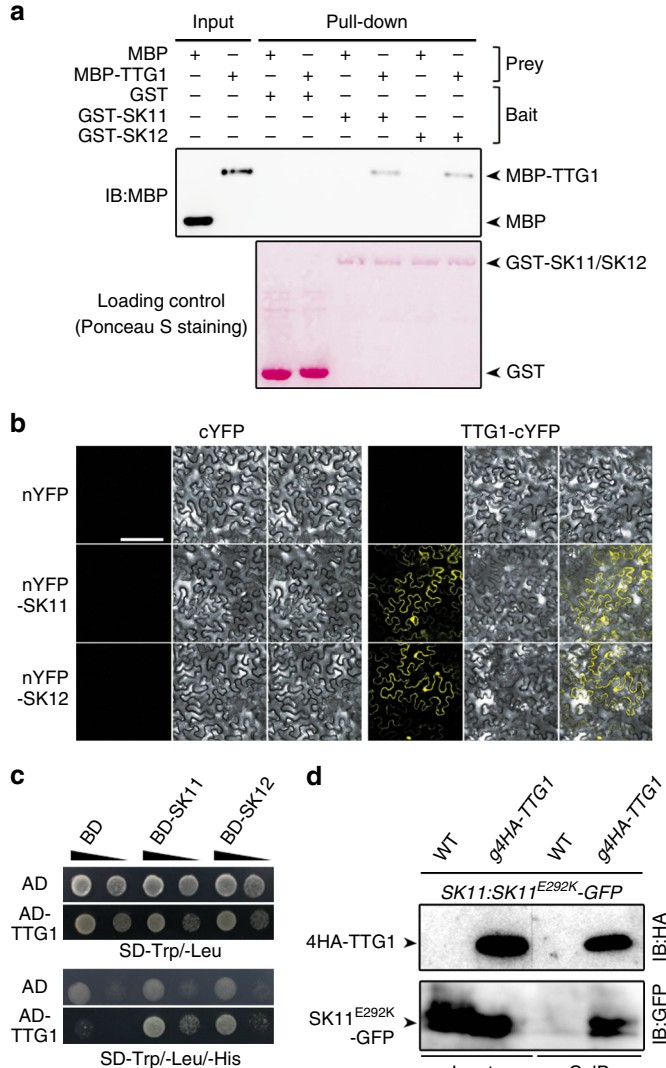

**Fig. 1** SK11 and SK12 interact with TTG1. **a** GST pull-down assay of the interaction of TTG1 with SK11 and SK12. GST, GST-SK11, and GST-SK12 were used as baits, and their loading amounts stained with Ponceau S are shown in the bottom panel. The input of prey proteins, MBP and MBP-TTG1, and their corresponding pull-downed signals are shown in the upper panel. **b** BiFC analysis of the interaction between TTG1 and SK11/SK12 in tobacco epidermal cells. Scale bar, 200 μm. **c** Yeast two-hybrid assay of the interaction between TTG1 and SK11/SK12. Ten-fold serial dilutions of transformed yeast cells were grown on SD-Trp/-Leu medium (upper panel) and SD-Trp/-Leu/-His medium supplemented with 0.3 mM 3-amino-1, 2, 4-triazole (3AT) (lower panel). **d** In vivo interaction between 4HA-TTG1 and SK11$^{E292K}$-GFP in *Arabidopsis* siliques shown by co-immunoprecipitation (CoIP). Total protein extracts from siliques of *SK11: SK11$^{E292K}$-GFP* in wild-type (WT) or *g4HA-TTG1* background 4 days after pollination (DAP) were immunoprecipitated by anti-HA antibody. The input and co-immunoprecipitated proteins were subjected to immunoblot analysis with anti-HA (upper panel) or anti-GFP antibody (lower panel). Uncropped original scans of immunoblots are shown in Supplementary Fig. 17

gain-of-function mutation from glutamic acid (E) to lysine (K) within the conserved TREE-motif in GSK3-like kinases, which resulted in a higher protein stability of SK11$^{E292K}$-GFP than SK11-GFP (Supplementary Fig. 5a) as demonstrated by several other GSK3-like kinases[28–31]. One representative *SK11: SK11$^{E292K}$-GFP* line (#5-1) that could contain only one T-DNA

insertion site based on the segregation ratio was selected for further investigation. We further generated *g4HA-TTG1 SK11: SK11$^{E292K}$-GFP* by crossing representative *ttg1-13 g4HA-TTG1* (#1-1) and *SK11:SK11$^{E292K}$-GFP* (#5-1) lines, and detected in vivo interaction between 4HA-TTG1 and SK11$^{E292K}$-GFP by co-immunoprecipitation analysis on protein extracts from *g4HA-TTG1 SK11:SK11$^{E292K}$-GFP* siliques 4 days after pollination (Fig. 1d). These results suggest that TTG1 interacts with SK11 and SK12 during seed development.

**SK11 and SK12 are highly expressed in young seeds**. To examine the global expression levels of *SK11* and *SK12*, we performed quantitative real-time PCR analysis of total RNA extracted from various aerial tissues of wild-type plants. *SK11* was expressed in all the tissues examined with the highest expression in young siliques collected before 7 days after pollination (Fig. 2a), while *SK12* expression was much higher in rosette leaves and young siliques than other aerial tissues (Fig. 2b). The expression levels of *SK11* and *SK12* were generally unaltered in aerial tissues of *ttg1-13* or *tt2-5* compared to wild-type plants (Supplementary Fig. 6a–d). *SK11* and *SK12* exhibited a similar trend of temporal expression in developing wild-type seeds (Fig. 2c, d). Expression of both genes remained at high levels from early to mid-embryo morphogenesis phase (0–4 days after pollination), but progressively decreased at the seed maturation phase. This pattern was comparable to the temporal expression pattern of TTG1, which peaked in seeds 4 days after pollination, but decreased afterwards (Supplementary Fig. 6e).

To monitor the detailed expression patterns of *SK11* and *SK12*, we transcriptionally fused the 2.6-kb and 2.3-kb 5′ upstream sequences of *SK11* and *SK12*, respectively, to the *β-glucuronidase* (GUS) reporter gene. Most transgenic lines generated for *SK11: GUS* or *SK12:GUS* displayed similar GUS staining patterns. GUS signals in *SK11:GUS* and *SK12:GUS* were detectable in most vegetative and reproductive tissues examined, and were relatively strong in some tissues, such as rosette leaves and radicle tips of mature embryos (Supplementary Fig. 7a). Furthermore, both *SK11:GUS* and *SK12:GUS* displayed staining signals in whole developing seeds 2 days after pollination (Fig. 2e, left panels; Supplementary Fig. 7b). In seeds 4 days after pollination, the strongest signals of *SK11:GUS* and *SK12:GUS* were found in the chalaza, while less strong, but evident signals were in the integument except the outmost layer that is being specialized for accumulation of mucilage constituents and starch granules (Fig. 2e, right panels). These GUS staining patterns in seeds are generally consistent with the expression data from the *Arabidopsis* eFP browser[32–34] (Fig. 2e and Supplementary Fig. 7c). Notably, robust expression of *SK11* and *SK12* in maternal tissues of seeds, such as the chalaza and integument, overlapped with strong *TTG1* expression in these tissues (Supplementary Fig. 7c).

**TTG1 and SK11/SK12 antagonistically affect seed characters**. To investigate the biological function of *SK11* and *SK12*, we isolated two corresponding T-DNA insertional mutants, *sk11* (SALK_014382) and *sk12* (CS332559), respectively (Supplementary Fig. 8a). There was no detectable expression of *SK11* and *SK12* in *sk11* and *sk12*, respectively (Supplementary Fig. 8b), implying that both lines could be null mutants. As *sk11* and *sk12* single mutants exhibited indistinguishable phenotypes from wild-type plants, we further created *sk11 sk12* double mutants since these two genes were highly similar.

Examination of seeds of *sk11 sk12* and the *SK11* gain-of-function transgenic line *SK11:SK11$^{E292K}$-GFP* (#5-1) revealed several seed coat defects associated with SK activities (Fig. 3a). In the wild-type seed coat, the outer cell layer of the outer

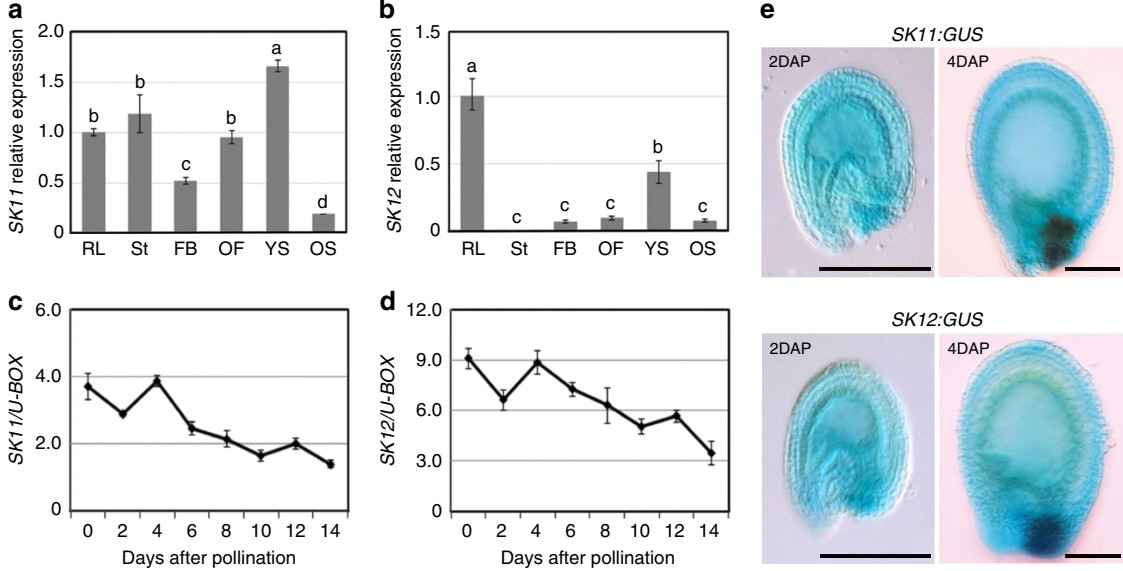

**Fig. 2** Expression patterns of *SK11* and *SK12*. **a**, **b** Quantitative real-time PCR analysis of *SK11* (**a**) and *SK12* (**b**) expression in aerial tissues of adult wild-type plants. RL, rosette leaf; St, stem; FB, flower bud; OF, open flower; YS, young silique (before 7 days after pollination); OS, old silique (7–12 days after pollination). Expression values normalized against the expression levels of *U-BOX* are shown relative to the level in rosette leaves set as 1. Values are mean ± s.d. of three biological replicates. Samples in each panel indicated with the same letter (a–d) above the bars are not significantly different. *P* values were determined by global one-way ANOVA with Holm-Bonferroni post hoc tests ($\alpha = 0.05$). **c**, **d** Quantitative real-time PCR analysis of *SK11* (**c**) and *SK12* (**d**) expression in developing seeds of wild-type plants. Results were normalized against the expression levels of *U-BOX* as an internal control. Values are mean ± s.d. of three biological replicates. **e** Representative GUS staining of *SK11:GUS* and *SK12:GUS* seeds 2 (left panels) and 4 (right panels) days after pollination (DAP). Scale bars, 200 μm

integument deposits mucilage mainly consisting of acidic polysaccharides around a polarized secondary cell wall forming a volcano-shaped structure called the columella[16,35] (Fig. 3a). During seed imbibition, the mucilage is extruded and forms a gelatin-like coating surrounding the seed, which may facilitate seed hydration and balance seed dormancy and germination in response to environmental conditions. We found that *sk11 sk12* developed larger columellae on the seed coat than wild-type plants, whereas *SK11:SK11^E292K^-GFP* seeds had smaller columellae (Fig. 3a and Supplementary Fig. 5c). As the formation of columella is associated with mucilage deposition on the seed coat[16], we also observed denser or sparser mucilage extrusion on the seed coat of *sk11 sk12* or *SK11:SK11^E292K^-GFP*, respectively, as compared with wild-type plants (Fig. 3a and Supplementary Fig. 5c). In addition, unstained seeds of *sk11 sk12* or *SK11:SK11^E292K^-GFP* were darker or paler than wild-type seeds, respectively (Fig. 3a and Supplementary Fig. 5c). In agreement with this observation, staining of seeds by *p*-dimethylaminocinnamaldehyde (DMACA) that reacts with the seed coat pigments, such as PA polymers and their precursors[36], demonstrated higher or lower PA deposition on the seed coat of *sk11 sk12* or *SK11:SK11^E292K^-GFP* than wild-type plants (Fig. 3a and Supplementary Fig. 5c). As expected, the gain-of-function *SK11:SK11^E292K^-GFP* line suppressed seed coat phenotypes of *sk11 sk12* (Supplementary Fig. 5c). Meanwhile, like other *ttg* mutants[5,9,11,15], *ttg1-13* exhibited several seed coat defects, such as loss of columella and mucilage, transparent testa, and reduced PA deposition (Fig. 3a). Taken together, these results demonstrate that *SK11:SK11^E292K^-GFP* partially phenocopies *ttg1-13*, whereas *sk11 sk12* exhibits opposite seed coat phenotypes.

As TTG1 also regulates the accumulation of seed storage reserves including fatty acids[17], we further examined whether *SK11* and *SK12* affect fatty acid contents in mature seeds. Although *SK11* and *SK12* barely affected the oil fatty acid composition in seeds (Supplementary Fig. 9a), total fatty acid levels in seeds of *sk11* and *sk12* single mutants were reduced

compared to those in wild-type seeds, and the reduction was more remarkable in *sk11 sk12* (Fig. 3b). In contrast, the levels of fatty acids in *SK11:SK11^E292K^-GFP* in either the wild-type or *sk11 sk12* background were significantly higher (Fig. 3b). These results, together with the negative effect of TTG1 on fatty acid levels in seeds[17], indicate that TTG1 and SK11/SK12 also play an antagonistic role in controlling fatty acid levels in seeds. Consistently, a group of genes involved in synthesis of fatty acids were downregulated in developing *sk11 sk12* seeds (Fig. 3c and Supplementary Fig. 9b), whereas most of these genes were found to be upregulated in *ttg1* developing seeds[17].

**TTG1 and TT2 act downstream of SK11/SK12.** As *TTG1 (At5g24520)* and *SK11 (At5g26751)* are closely linked on the chromosome 5, we were unable to create their double mutants through genetic crossing. Instead, we applied treatment using bikinin, which specifically inhibits subgroups I and II GSK3-like kinases[37], to test the genetic interaction between *TTG1* and *SK11/SK12*. Bikinin treatment of wild-type siliques 4 days after pollination resulted in darker seed coat and less fatty acids than mock treatment (Fig. 3d, g), which are in accordance with *sk11 sk12* seed phenotypes (Fig. 3a). In contrast, Bikinin treatment almost had no effect on seed coat color and fatty acid levels of *ttg1-13* (Fig. 3e, g), indicating that TTG1 acts downstream of *SK11* and *SK12*.

We then explored the genetic interaction between SK11/SK12 and two potential TTG1 interacting partners, TT2 and TT8, because they also affect seed coat color and fatty acid synthesis[38–41]. We first compared temporal expression of *TT2* and *TT8* in developing seeds and the seed coat phenotypes of their corresponding mutants. *TT2* exhibited a similar temporal expression pattern to *TTG1* in developing seeds, which peaked at the embryo morphogenesis phase (4 days after pollination), but subsequently decreased (Supplementary Fig. 6e, f), whereas *TT8* peaked at the seed maturation phase (8 days after pollination)

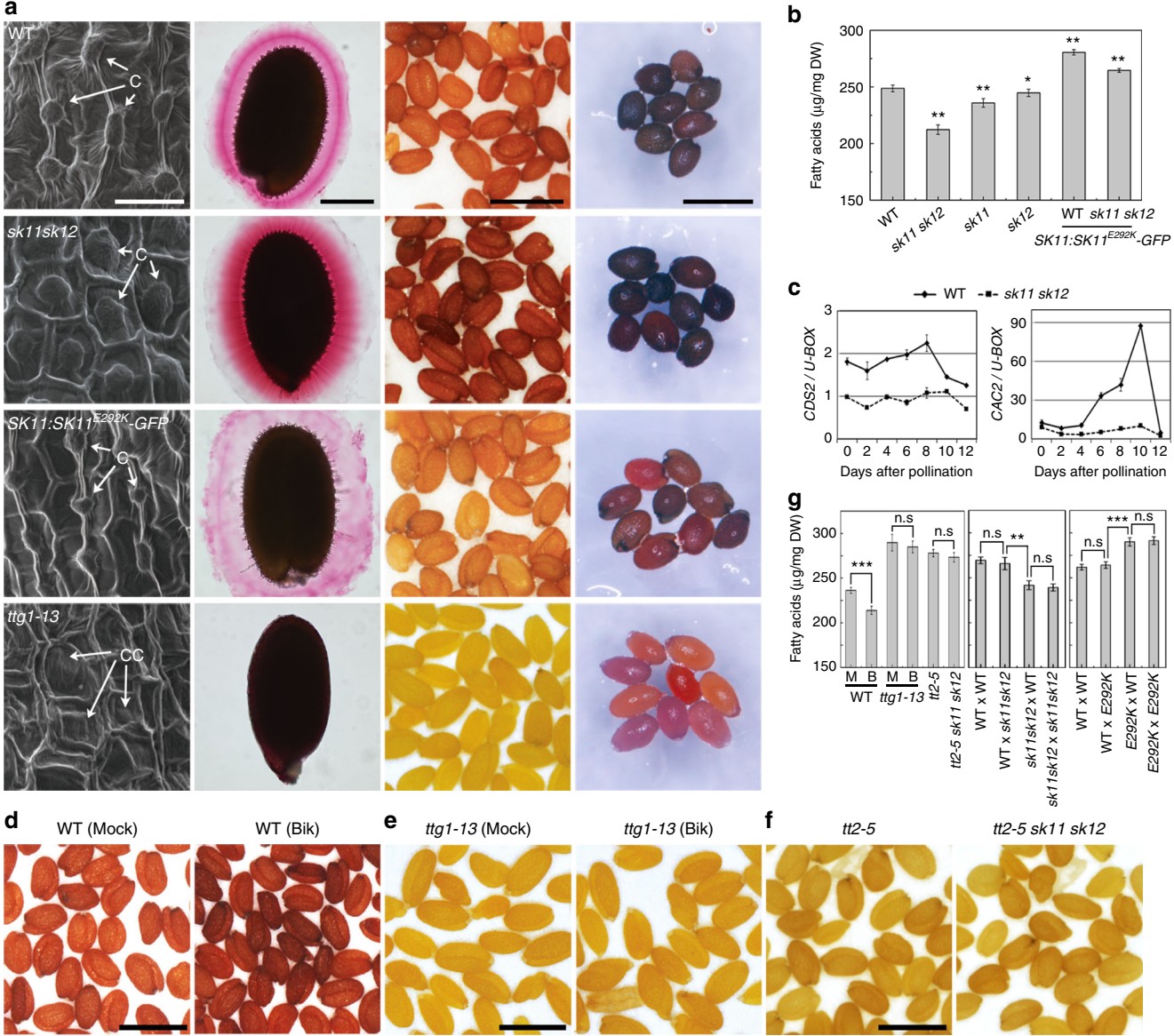

**Fig. 3** Altered activity of SKs affect seed characters. **a** Examination of seed coat phenotypes of mature *Arabidopsis* seeds in various genetic backgrounds. Scanning electron microscopy (SEM) of seed coat (first column), seed coat mucilage staining with ruthenium red (second column), seed color (third column), and seed staining with *p*-dimethylaminocinnamaldehyde (DMACA) (fourth column) are shown in panels from left to right. Scale bars, 25 μm, 200 μm, 1 mm, and 1 mm (from left to right). In SEM panels, "C" indicates columella, while "CC" indicates collapsed columella. **b** Measurement of total fatty acid contents of mature seeds in various genetic backgrounds. Values are mean ± s.d. of three biological replicates. Asterisks indicate significant differences between wild-type and samples in other genetic backgrounds (two-tailed paired Student's *t* test, *$P < 0.05$, ***$P < 0.001$). **c** Quantitative real-time PCR analysis of *CDS2* and *CAC2* expression in developing seeds of wild-type and *sk11 sk12* plants. Results were normalized against the expression levels of *U-BOX* as an internal control. Values are mean ± s.d. of three biological replicates. **d**, **e** Effect of bikinin treatment on seed color of wild-type (**d**) and *ttg1-13* (**e**) mature seeds. Wild-type and *ttg1-13* siliques were mock-treated (Mock) or treated with 25 μM bikinin (Bik) at 4 days after pollination, and the resulting mature seeds were collected for further analyses. Scale bars, 1 mm. **f** Comparison of seed color of *tt2-5* and *tt2-5 sk11 sk12*. Scale bar, 1 mm. **g** Measurement of total fatty acid contents of mature seeds described in **d**–**f** (left panel) and mature F1 seeds from reciprocal crosses between wild-type and *sk11 sk12* plants (middle panel) or between wild-type and *SK11:SK11^{E292K}-GFP* (*E292K*) plants (right panel). Values are mean ± s.d. of three biological replicates. Asterisks indicate significant differences between each specified pair of samples in different genetic backgrounds (two-tailed paired Student's *t* test, **$P < 0.01$, ***$P < 0.001$, no statistical difference (n.s), $P > 0.05$). M, mock-treated; B, bikinin-treated

(Supplementary Fig. 6g). In addition, although both *tt2-5* and *tt8-4* seeds displayed transparent testa and reduced PA deposition, only *tt2-5* exhibited the defect in mucilage extrusion (Supplementary Fig. 10) as observed in *SK11:SK11^{E292K}-GFP* (Fig. 3a). These results imply that TT2 could be more closely related to TTG1 activity in affecting seed characters. We then created *tt2-5 sk11 sk12* triple mutants, and found that *tt2-5* completely suppressed *sk11 sk12* phenotypes pertaining to seed coat color

(Fig. 3f) and fatty acid levels (Fig. 3g, left panel). These observations suggest that both TTG1 and its interacting partner TT2 control seed characters genetically downstream of *SK11* and *SK12*.

While *SK11* and *SK12* affect the phenotypes of seed coat that is a maternal tissue, we also tested the origin of their effects on fatty acid levels in seeds. Reciprocal crosses between wild-type and *sk11 sk12* plants (Fig. 3g, middle panel) and between wild-type

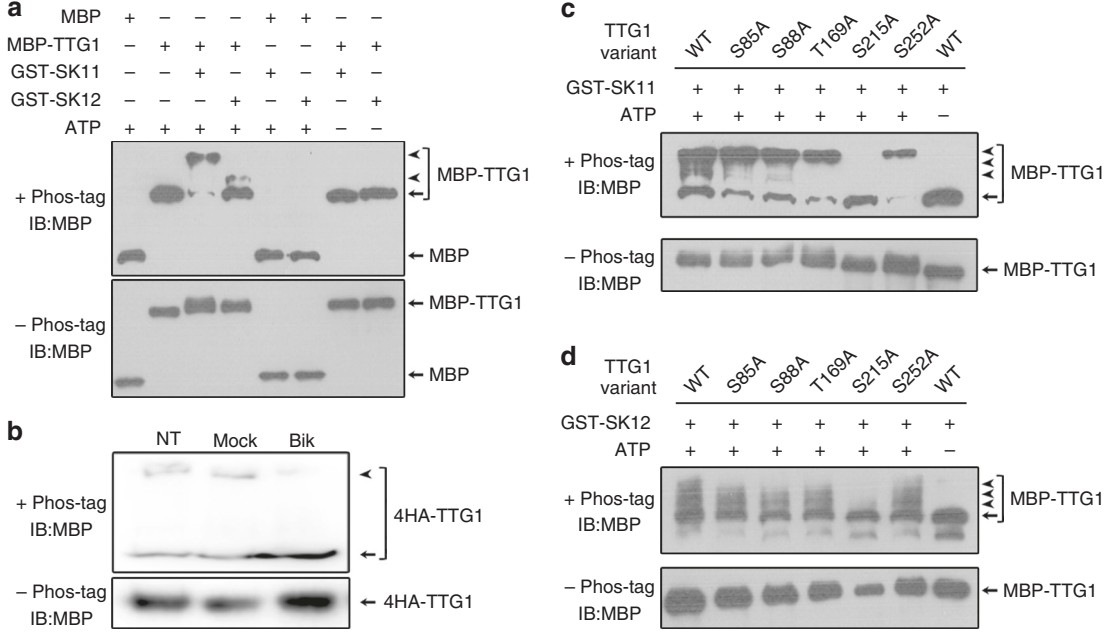

**Fig. 4** SK11 and SK12 phosphorylate TTG1 mainly through Ser 215. **a** GST-SK11 and GST-SK12 alter the electrophoretic mobility of MBP-TTG1 on a Phos-tag-containing gel (upper panel), but not on a regular SDS-PAGE gel (lower panel). GST-SK11 and GST-SK12 do not affect the electrophoretic mobility of MBP on both Phos-tag-containing and regular SDS-PAGE gels. Arrowheads and arrows indicate phosphorylated and unphosphorylated protein forms, respectively. **b** Bikinin treatment inhibits phosphorylation of 4HA-TTG1 in *ttg1-13 g4HA-TTG1*. Total protein was extracted from *ttg1-13 g4HA-TTG1* siliques collected at 4 days after pollination, which were not treated (NT), mock-treated (Mock), or treated by bikinin (Bik) for 2 h. These protein extracts were analyzed by immunoblot on Phos-tag-containing (upper panel) and regular SDS-PAGE (lower panel) gels using anti-MBP antibody. Arrowhead and arrow indicate phosphorylated and unphosphorylated 4HA-TTG1, respectively. **c, d** Immunoblot analysis of phosphorylation of MBP-TTG1 variants containing single site mutations at serine or threonine by GST-SK11 (**c**) or GST-SK12 (**d**) on Phos-tag-containing (upper panels) and regular SDS-PAGE (lower panels) gels. In vitro phosphorylation of wild-type TTG1 (MBP-TTG1) with or without ATP serves as a positive or negative control, respectively. Arrowheads and arrows indicate phosphorylated and unphosphorylated MBP-TTG1, respectively. Uncropped original scans of immunoblots are shown in Supplementary Fig. 17

and *SK11:SK11^{E292K}-GFP* (Fig. 3g, right panel) both revealed a maternal effect of SK11 or SK12 on fatty acid levels in seeds, indicating that SK11 and SK12 maternally regulate seed characters.

**SK11 and SK12 phosphorylate TTG1 in vitro and in vivo.** Since SK11 and SK12 interacted with TTG1 and affected seed characters upstream of TTG1, we reasoned that TTG1 could serve as a substrate of SK11 and SK12. To this end, we performed in vitro kinase assay using Phos-tag SDS-PAGE, and detected a mobility shift of phosphorylated MBP-TTG1 by GST-SK11 and GST-SK12 in the phosphate-affinity SDS-PAGE (Fig. 4a). More importantly, we also observed a mobility shift of phosphorylated 4HA-TTG1 from protein extracts of *ttg1-13 g4HA-TTG1* siliques using Phos-tag SDS-PAGE, whereas phosphorylation of 4HA-TTG1 was inhibited in protein extracts from bikinin-treated siliques (Fig. 4b), suggesting that TTG1 is indeed phosphorylated in vivo by GSK3-like kinases.

To identify specific phosphorylation sites on TTG1 mediated by SK11 and SK12, we analyzed the phosphorylated MBP-TTG1 products purified from the in vitro kinase assay by LC-MS/MS, and found five potential phosphorylation sites, including Ser 85, Ser 88, threonine (Thr) 169, Ser 215, and Ser 252 (Supplementary Fig. 11a–e). We further examined these five sites by in vitro kinase assays, and revealed that only mutation of Ser (S) 215 to alanine (A) greatly abolished in vitro phosphorylation of MBP-TTG1 by both GST-SK11 and GST-SK12 (Fig. 4c, d and Supplementary Fig. 11d, e), indicating that Ser 215 targeted by SK11 and SK12 is essential for phosphorylation of TTG1.

**Phosphorylation of TTG1 at Ser 215 affects seed characters.** To investigate the biological effects of TTG1 phosphorylation, we transformed *ttg1-13* with a phosphorylation-mimicking construct *g4HA-TTG1^{S215E}* or a dephosphorylation-mimicking construct *g4HA-TTG1^{S215A}* (Fig. 5a), in which Ser 215 was mutated to glutamic acid (E) or alanine (A), respectively, in the *g4HA-TTG1* genomic fragment that was able to fully rescue *ttg1-13* as early described (Fig. 1d, Supplementary Fig. 4). We obtained 12 *ttg1-13 g4HA-TTG1^{S215E}* and 15 *ttg1-13 g4HA-TTG1^{S215A}* independent lines. Since most lines for each genotype showed comparable phenotypes (Supplementary Fig. 12), we selected one representative line each (*ttg1-13 g4HA-TTG1^{S215E} #2-1* and *ttg1-13 g4HA-TTG1^{S215A} #1-1*) with a putative single-copy transgene based on a 3:1 segregation ratio for further detailed comparison of phenotypes among plants with various genetic backgrounds.

*g4HA-TTG1* was able to fully rescue the pleiotropic defects of *ttg1-13*, including glabrous leaves (Supplementary Fig. 13a), altered pattern of root hair formation (Supplementary Fig. 13b), and abnormal seed characters (Fig. 5b, c and Supplementary Fig. 4), such as loss of columellae and mucilage, transparent testa, reduced PA deposition, and increased fatty acid levels. Like *g4HA-TTG1*, *g4HA-TTG1^{S215E}* and *g4HA-TTG1^{S215A}* could equally rescue leaf and root hair phenotypes of *ttg1-13* (Supplementary Fig. 13a, b), indicating that phosphorylation of TTG1 may not affect TTG1 function in mediating epidermal morphogenesis at the post-embryonic stage.

However, *g4HA-TTG1^{S215E}* and *g4HA-TTG1^{S215A}* had opposite effects on seed characters of *ttg1-13* (Fig. 5b, c). *ttg1-13 g4HA-TTG1^{S215E}* partially phenocopied *SK11:SK11^{E292K}-GFP*, showing loss of columellae and reduced mucilage extrusion, pale seeds,

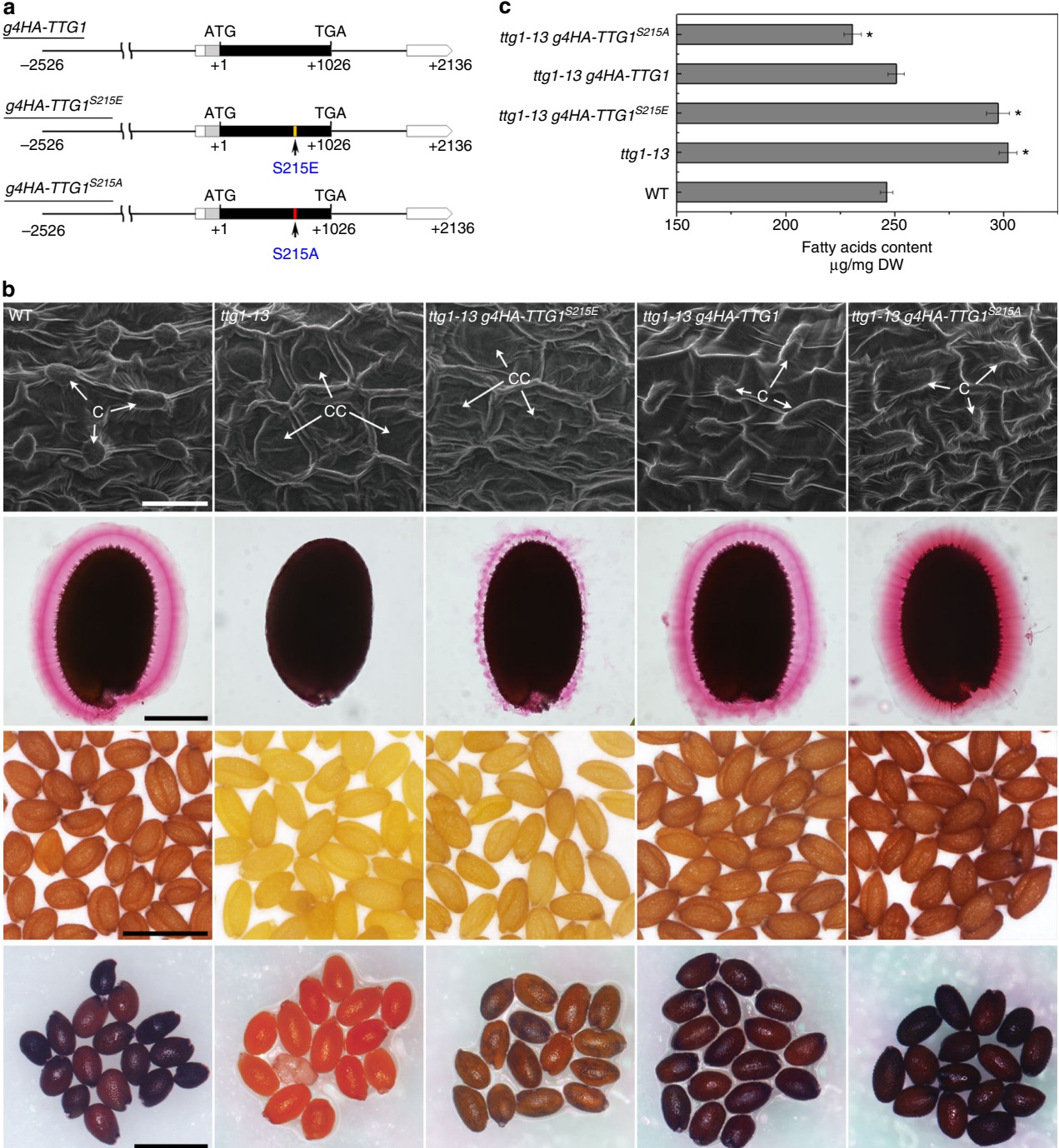

**Fig. 5** Phosphorylation of TTG1 at Ser 215 abolishes its activity in regulating seed characters. **a** Schematic diagram of the phosphorylation-mimicking construct *g4HA-TTG1^S215E^* and the dephosphorylation-mimicking construct *g4HA-TTG1^S215A^*. The coding and untranslated regions of *TTG1* are indicated by black and white boxes, respectively, while an intron and other genomic regions are indicated by black lines. The 4HA tag represented by a gray box is fused to the N terminus of TTG1. The first nucleotide of the translation start codon is assigned the +1 position, and other sequences are numbered relative to this site. Arrow indicates the site containing a point mutation from S to E or S to A at Ser 215. **b** Examination of seed coat phenotypes of mature *Arabidopsis* seeds in various genetic backgrounds. SEM of seed coat (first row), seed coat mucilage staining with ruthenium red (second row), seed color (third row), and seed staining with DMACA (fourth row) are shown in panels from top to bottom. Scale bars, 25 μm, 200 μm, 1 mm, and 1 mm (from top to bottom). In SEM panels, "C" indicates columella, while "CC" indicates collapsed columella. **c** Measurement of total fatty acid contents of mature seeds in various genetic backgrounds. Values are mean ± s.d. of three biological replicates. Asterisks indicate significant differences between wild-type and samples in other genetic backgrounds (two-tailed paired Student's *t* test, $P < 0.001$)

reduced PA deposition, and significantly increased fatty acid levels compared to wild-type seeds, whereas *ttg1-13 g4HA-TTG1^S215A^* displayed opposite seed phenotypes, resembling those of *sk11 sk12* (Figs. 3a and 5b, c). These results demonstrate that

phosphorylation of TTG1 at Ser 215, which is conserved among TTG1 orthologs in various plant species (Supplementary Fig. 14a, b), by SK11 and SK12 inhibits formation of seed mucilage and flavonoid pigments, but promotes synthesis of fatty acids,

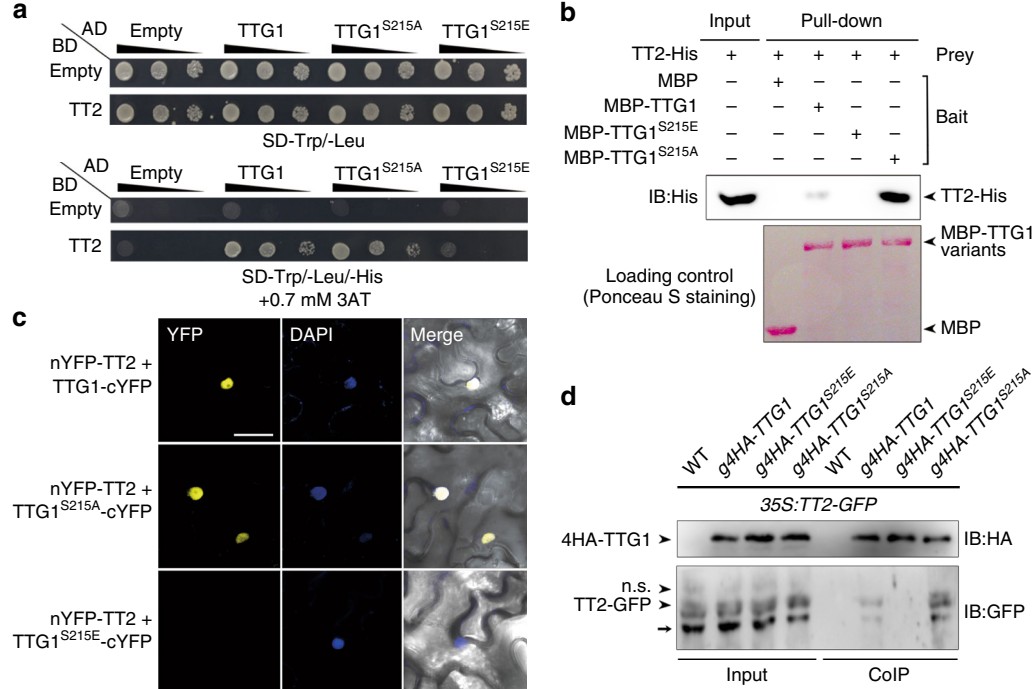

**Fig. 6** Phosphorylation of TTG1 at Ser 215 abolishes its interaction with TT2. **a** Yeast two-hybrid assays show the interaction of TT2 with TTG1, its phosphorylation-mimicking mutant TTG1$^{S215E}$ and dephosphorylation-mimicking mutant TTG1$^{S215A}$. Ten-fold serial dilutions of transformed yeast cells were grown on SD-Trp/-Leu medium (upper panel) and SD-Trp/-Leu/-His medium supplemented with 0.7 mM 3AT (lower panel). **b** In vitro pull-down assay of the interaction of TT2 with TTG1, TTG1$^{S215E}$, and TTG1$^{S215A}$. MBP, MBP-TTG1, MBP-TTG1$^{S215E}$, and MBP-TTG1$^{S215A}$ were used as baits, and their loading amounts stained with Ponceau S are shown in the bottom panel. The input of the prey protein TT2-His and its corresponding pull-downed signals are shown in the upper panel. **c** BiFC analysis of the interaction of TT2 with TTG1, TTG1$^{S215E}$, and TTG1$^{S215A}$ in tobacco epidermal cells. Yellow fluorescence protein (YFP), fluorescence of yellow fluorescent protein; DAPI, fluorescence of 4′,6-diamino-2-phenylindol; Merge, merge of YFP, DAPI, and bright field. Scale bar, 50 μm. **d** In vivo interaction between 4HA-TTG1 and TT2-GFP in *Arabidopsis* siliques shown by co-immunoprecipitation (CoIP). Total protein extracts from siliques of *35S:TT2-GFP* in wild-type (WT), *g4HA-TTG1*, *g4HA-TTG1$^{S215E}$*, or *g4HA-TTG1$^{S215A}$* background 4 days after pollination (DAP) were immunoprecipitated by anti-HA antibody. The input and co-immunoprecipitated proteins were subjected to immunoblot analysis with anti-HA (upper panel) or anti-GFP antibody (lower panel). The non-specific (n.s.) signal and the potential degraded form of TT2-GFP are indicated by an arrowhead and an arrow, respectively. Uncropped original scans of immunoblots are shown in Supplementary Fig. 17

whereas unphosphorylated TTG1 has an opposite effect on these seed characters. As the seed coat defects of *ttg1-13* was still slightly rescued by *g4HA-TTG1$^{S215E}$* (Fig. 5b and Supplementary Fig. 12c), it is possible that TTG1 function in the seed coat is not fully phosphorylation-dependent.

**Phosphoserine 215 of TTG1 abolishes interaction with TT2.** The findings on the similar functions of TTG1 and TT2 in controlling seed characters downstream of *SK11* and *SK12* (refs. [9,15,17,38,40]) (Fig. 3e, f, g and Supplementary Fig. 10), together with the reported in vitro protein interaction between TTG1 and TT2 (ref. [18]) and their similar temporal mRNA expression pattern in developing seeds (Supplementary Fig. 6e, f), prompted us to investigate whether different effects of TTG1 phosphorylation and unphosphorylation at Ser 215 on seed characters are relevant to altered TTG1 interaction with TT2.

Yeast two-hybrid assays using serial dilutions of transformed yeast cells revealed that TTG1$^{S215E}$ failed to interact with TT2, while TTG1 and TTG1$^{S215A}$ interacted with TT2 to the same extent (Fig. 6a). In vitro pull-down assays demonstrated that TT2-His had a stronger interaction with MBP-TTG1$^{S215A}$ than MBP-TTG1, but did not interact with MBP-TTG1$^{S215E}$ (Fig. 6b). As MBP-TTG1 was not phosphorylated in our bacterial expression system (Fig. 4a), a stronger interaction between TT2-His and MBP-TTG1$^{S215A}$ in pull-down assays may be due to the effect of S215A mutation on other biochemical characteristics

of TTG1 rather than phosphorylation of TTG1. In addition, we also did not detect the interaction between TT2 and TTG1$^{S215E}$ in BiFC assays (Fig. 6c). These results all indicate that the phosphorylated TTG1 at Ser 215 is unable to interact with TT2. To test this possibility in vivo, we generated *35S:TT2-GFP* in wild-type, *g4HA-TTG1*, *g4HA-TTG1$^{S215A}$*, and *g4HA-TTG1$^{S215E}$* backgrounds, and performed co-immunoprecipitation analysis on protein extracts from their siliques 4 days after pollination. Indeed, TT2-GFP interacted with 4HA-TTG1$^{S215A}$ stronger than 4HA-TTG1, but not with 4HA-TTG1$^{S215E}$ (Fig. 6d), confirming that phosphorylation of TTG1 at Ser 215 abolishes the interaction between TTG1 and TT2 in siliques. We further examined the seed phenotypes of *tt2-5 ttg1-13 g4HA-TTG1$^{S215A}$*, and found that this plant exhibited similar seed phenotypes to *tt2-5* (Supplementary Fig. 15), substantiating that TT2 is important for mediating TTG1 effect on seed phenotypes.

**Phosphoserine 215 of TTG1 weakens TTG1 binding to GL2.** To find out how interaction between TTG1 and TT2 affects downstream molecules to control seed characters, we selected *GL2* as a potential target of TTG1 and TT2 based on the following two reasons. First, loss of GL2 activity results in higher levels of seed oil, deficiency in seed mucilage production, collapsed columella, light seed color and slightly reduced PA deposition[3,22–24] (Supplementary Fig. 16), which are similar to those observed in *ttg1* and *tt2* mutants[9,15,17,38,40] (Fig. 3a and Supplementary Fig. 10).

Second, it has been suggested that *GL2* could be a direct target of TTG1 in trichome patterning[21], and also serves as a downstream target of TT2 in affecting fatty acid biosynthesis in seeds[40].

We then performed chromatin immunoprecipitation (ChIP) assays using siliques of *ttg1-13 g4HA-TTG1* and *35S::TT2-GFP* to test whether TTG1 and TT2 are directly associated with the *GL2* locus in seed development, respectively (Fig. 7a–c). Chromatin immunoprecipitation assays showed that both 4HA-TTG1 and TT2-GFP were strongly associated with the transcription start site near the number 8 and 9 fragments with the highest enrichment fold (Fig. 7b, c). This observation, together with the interaction between TTG1 and TT2 (Fig. 6), implies that TTG1 and TT2 interact to physically bind to the same *GL2* genomic region. We further found that 4HA-TTG1$^{S215E}$, which prevented the interaction between TTG1 and TT2 (Fig. 6), also weakened TTG1 binding to the *GL2* locus (Fig. 7b). Consistently, 4HA-TTG1 binding to *GL2* was significantly weaker in *tt2-5* than in the wild-type background (Fig. 7b). In contrast, TT2-GFP binding to *GL2* in *ttg1-13* and wild-type backgrounds remained the same (Fig. 7c). Thus, while TT2 binding to *GL2* is independent of TTG1, TTG1 binding to *GL2* partially relies on its interaction with TT2, which is mediated by the TTG1 phosphorylation status at Ser 215.

Notably, the *GL2* genomic region near the fragment 9 contained an AC-rich motif that is typically recognized by MYB transcription factors[42], such as TT2 (Fig. 7d). To confirm our ChIP assay results, we then carried out the electrophoretic mobility shift assay (EMSA) using the biotinylated probe containing this AC-rich motif or its mutated form. Electrophoretic mobility shift assay showed that MBP-TT2 only bound to the native probe, which was specifically eliminated by native cold competitors, whereas MBP-TTG1 itself did not bind to the native probe (Fig. 7e). However, in the presence of MBP-TT2, MBP-TTG1 was able to bind to the native probe as revealed by observable supershifts because of larger complexes composed of MBP-TT2, MBP-TTG1, and the native probe (Fig. 7f). Taken together, ChIP and EMSA assays suggest that recruitment of TTG1 to the *GL2* locus is at least partially dependent on its interaction with TT2.

Quantitative real-time PCR analysis revealed that in developing seeds, *GL2* expression gradually increased at the embryo morphogenesis phase, and peaked at 6 days after pollination before seed maturation started (Supplementary Fig. 6h). Its expression peak was slightly delayed compared to those of *TTG1* and *TT2* (Supplementary Fig. 6e, f, h). We further measured *GL2* expression in siliques in various genetic backgrounds 4 days after pollination, and found that *GL2* was downregulated in both *ttg1-13* and *tt2-5* (Fig. 7g). Its expression was recovered to the wild-type level in *ttg1-13 g4HA-TTG1*, but remained low in *ttg1-13 g4HA-TTG1$^{S215E}$*, supporting that the TTG1 phosphorylation status at Ser 215 ultimately affects *GL2* expression. Consistently, as SK11 and SK12 mediate TTG1 phosphorylation at Ser 215, *GL2* was upregulated in *sk11 sk12*, but downregulated in *SK11:SK11$^{E292K}$-GFP*. Similarly, bikinin treatment, which inhibits SK11 and SK12, upregulated *GL2* in wild-type siliques (Fig. 7g). As upregulation of *GL2* by bikinin treatment was much stronger than that in *sk11 sk12*, it is possible that bikini could affect other subgroups I and II GSK3-like kinases, which may share a similar role of SK11 and SK12 in affecting *GL2* expression.

## Discussion

During seed development, sucrose is the major form of photosynthetically assimilated carbon source that is transported from the parent plant to maternal tissues like integuments and filial tissues, such as embryos and endosperms. Sucrose, together with

other nutrients including amino acids and minerals from the parent plant, contributes to both the generation of the protective seed coat and the storage of reserve compounds in filial tissues for seed germination and subsequent seedling growth[1]. As supply of nutrients from the parent plants is usually limited, loading of nutrients into dominant sinks in developing seeds should be dynamically regulated so that the development of zygotic and maternal tissues can be coordinated to well prepare seeds for successful propagation of the species in changing environments. In this study, we have revealed a molecular framework in which site-specific phosphorylation of TTG1 serves as a master switch that specifically controls TTG1-mediated carbon partitioning among metabolic pathways in zygotic and maternal tissues in *Arabidopsis* seeds.

During seed development, TT2 preferably interacts with unphosphorylated TTG1 and facilitates TTG1 binding to the *GL2* locus to promote *GL2* expression (Fig. 8). In contrast, two GSK3-like kinases, SK11 and SK12, interact with TTG1 and specifically phosphorylate it at Ser 215, which compromises TTG1 interaction with TT2 and the consequent recruitment of TTG1 to the *GL2* locus, thus downregulating *GL2*. In agreement with this molecular framework, the gain-of-function plant *SK11:SK11$^{E292K}$-GFP*, *ttg1-13*, the phosphorylation-mimicking plant *ttg1-13 g4HA-TTG1$^{S215E}$*, *tt2-5*, and *gl2-3* all exhibit similar seed phenotypes, including decreased production of columellae and mucilage, pale seeds associated with reduced PA deposition, and increased fatty acid accumulation, although the severity of these phenotypes is different in these plants. On the contrary, *sk11 sk12* and the dephosphorylation-mimicking plant *ttg1-13 g4HA-TTG1$^{S215A}$* displayed opposite seed coat phenotypes and decreased fatty acid accumulation. Therefore, the site-specific phosphorylation of TTG1 at Ser 215 by SK11/SK12 is a molecular switch that promotes fatty acid synthesis in the embryo, but reduces production of mucilage and flavonoid pigments in the seed coat. A consistent observation in plant materials examined in this study is that changes in fatty acid levels in the embryo are reversely correlated with production of mucilage and flavonoid pigments in the seed coat. This is in agreement with the understanding that the metabolic pathways in producing fatty acids, flavonoid pigments (i.e., PAs and flavonols), and mucilage in seeds compete dynamically for the same carbon source, sucrose, from the parent plant[1,3,4]. Thus, phosphorylation or dephosphorylation of TTG1 at Ser 215 plays an important role in mediating carbon partitioning among these metabolic pathways during seed development.

Notably, reciprocal genetic crosses have revealed that SK11 and SK12 maternally regulate fatty acid levels (Fig. 3g). In addition, previous studies have shown the maternal effects of TTG1 and GL2 on fatty acid levels in seeds[3,17], and that *TT2* is predominantly expressed in the seed coat and controls seed coat development[38,43]. These results support that SK11/12-mediated TTG1 phosphorylation and it interaction with TT2 as well as their effects on *GL2* maternally regulate carbon partitioning in seeds. In addition, since all loss-of-function mutants discussed in this study still produce viable seeds, it seems that the defect in carbon partitioning during seed development might not directly affect seed viability under normal growth conditions.

By now, there have been very few studies of the signals upstream of SK11 and SK12. It has been reported that SK11 (also known as ASKα) is induced by abiotic stresses, such as salt stress, to regulate stress tolerance by activating glucose-6-phosphate dehydrogenase (G6PD) and affecting reactive oxygen species levels[44]. In addition, another study has shown that SK11 phosphorylates G6PD, thus constituting an immune signaling module in response to pathogenic microbes and linking protein phosphorylation cascades to metabolic regulation[45]. It is noteworthy

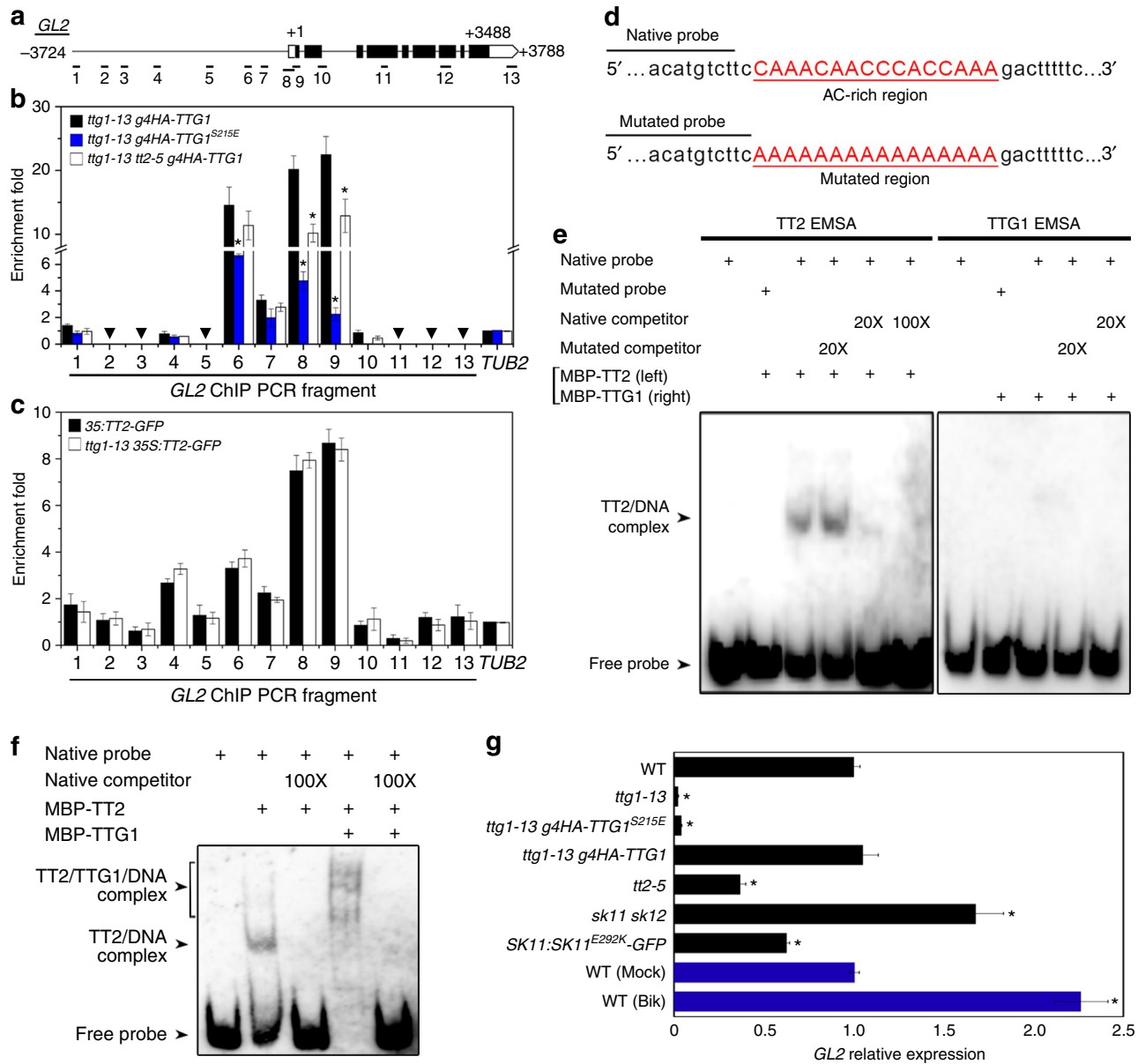

**Fig. 7** Phosphorylation of TTG1 at Ser 215 weakens TTG1 binding to the *GL2* locus through TT2. **a** Schematic diagram of the *GL2* genomic region. The coding and untranslated regions are indicated by black and white boxes, respectively, while introns and other genomic regions are indicated by black lines. The first nucleotide of the translation start codon is assigned the +1 position, and other sequences are numbered relative to this site. Thirteen DNA fragments (1–13) spanning the *GL2* locus were designed for ChIP analyses as shown in **b**, **c**. **b**, **c** ChIP analyses of binding of 4HA-TTG1 variants (**b**) and TT2-GFP (**c**) to the *GL2* locus. Chromatin immunoprecipitation analyses were performed on siliques in various genetic backgrounds 4 days after pollination. Nuclear extracts served as the input, while immunoprecipitated fractions by anti-HA antibody (**b**) and anti-GFP antibody (**c**) were used as the eluate. A *TUB2* fragment was amplified as a negative control. Triangles indicate barely detectable amplicons at certain sites. Values are mean ± s.d. of three biological replicates. Asterisks indicate significant differences in enrichment fold changes in comparison with *ttg1-13 g4HA-TTG1* (two-tailed paired Student's *t* test, *P* < 0.001). **d** List of the putative TT2-binding element (native probe) near primer 9 in the *GL2* genomic region shown in **a**, and its mutated version (mutated probe). The mutated AC-rich region is underlined in red. These probes were used for subsequent EMSA assays. **e**, **f** EMSA assays of binding of MBP-TT2 (**e**, left panel), MBP-TTG1 (**e**, right panel), and both proteins (**f**) to the probes shown in **c**. Recombinant proteins were incubated with biotinylated native or mutated probe. Nonlabeled native or mutated probes in 20- or 100-fold molar excess relative to the biotinylated native probe were used as cold competitors. Supershift complexes were observable (**f**) when MBP-TT2 and MBP-TTG1 were incubated with the biotinylated native probe. **g** Quantitative real-time PCR analysis of *GL2* expression in siliques in various genetic backgrounds 4 days after pollination. Wild-type siliques mock-treated (Mock) or treated with 25 μM bikinin (Bik) for 2 h were collected for expression analysis shown in blue bars. Expression values normalized against the expression levels of *U-BOX* are shown relative to the level in wild-type siliques set as 1. Values are mean ± s.d. of three biological replicates. Asterisks indicate significant differences in *GL2* expression between indicated genotypes and wild-type plants (black bars) or between bikinin- or mock-treated wild-type plants (blue bars) (two-tailed paired Student's *t* test, *P* < 0.001). Uncropped original scans of immunoblots are shown in Supplementary Fig. 17

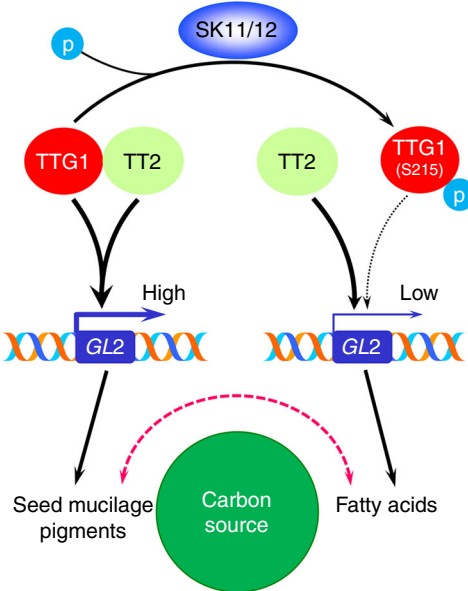

**Fig. 8** Site-specific phosphorylation of TTG1 by SK11/SK12 regulates carbon flow in *Arabidopsis* seeds. During seed development, TT2 interacts with unphosphorylated TTG1 and facilitates recruitment of TTG1 to the *GL2* locus, thus upregulating *GL2* expression. This promotes the formation of seed mucilage and flavonoid pigments. In contrast, two GSK3-like kinases, *SK11* and *SK12*, interact with TTG1 and specifically phosphorylate TTG1 at Ser 215. This abolishes TTG1 interaction with TT2, which compromises TTG1 binding to *GL2*, thus downregulating *GL2* expression to promote biosynthesis of fatty acids in seeds. Therefore, site-specific phosphorylation of TTG1 by SK11/SK12 specifically modulates the direction of carbon flow to either formation of seed mucilage and flavonoid pigments or synthesis of fatty acids via regulating *GL2* expression levels in *Arabidopsis* seeds

that G6PD catalyzes a key step of the oxidative pentose phosphate pathway that generates NADPH necessary for fatty acid biosynthesis. It will be interesting to investigate whether the link between SK11 and G6PD would influence the role of SK11 in targeting to another substrate TTG1 and the resulting effect on carbon partitioning in seeds.

Although TTG1 has been shown to affect various aspects of seed development and post-embryonic development in *Arabidopsis*[9–14], the site-specific phosphorylation of TTG1 at Ser 215 by SK11 and SK12 only specifically affects seed characters rather than other post-embryonic traits. *g4HA-TTG1* and its phosphorylation-mimicking mutant *g4HA-TTG1^S215E* or dephosphorylation-mimicking mutant *g4HA-TTG1^S215A* equally rescue post-embryonic phenotypes of *ttg1-13*, including glabrous leaves and altered pattern of root hair formation (Supplementary Fig. 13a, b), demonstrating that TTG1 function in regulating epidermal morphogenesis at the post-embryonic stage is insensitive to phosphorylation of TTG1 at Ser 215. This is in contrast to the opposite roles of *g4HA-TTG1^S215E* and *g4HA-TTG1^S215A* in specifically affecting *ttg1-13* seed characters (Fig. 5b, c). Similarly, while *sk11 sk12* and the gain-of-function plant *SK11:SK11^E292K*-*GFP* barely affect the formation of leaf trichome and root hair (Supplementary Fig. 13a, b), they display opposite seed characters in terms of production of fatty acids, mucilage, and flavonoid pigments (Fig. 3a, b). These observations, together with the result that *TTG1* genetically acts downstream of *SK11* and *SK12* (Fig. 3d, e, g), strongly suggest that phosphorylation of TTG1 at Ser 215 by SK11 and SK12 is a hitherto unknown mechanism that enables a specific tailoring of TTG1 functions in regulating the carbon flow among the metabolic pathways involved in seed development (Fig. 8). This finding potentially provides a new

means of targeted metabolic engineering of metabolites in seeds, such as oils and flavonoids, through post-translational modifications of key transcription factors, such as TTG1, which specifically affects desirable seed characters while maintaining their normal functions in regulating other plant traits.

In addition to the molecular framework described in this study, there is evidence implying that the regulatory hierarchy involving phosphorylation of TTG1 might engage other unknown mechanisms. First, LC-MS/MS analyses of in vitro phosphorylated MBP-TTG1 products have revealed several potential phosphorylation sites for SK11 or SK12 (Supplementary Fig. 11a–c). Although Ser 215 is the dominant and common site responsible for phosphorylation of TTG1 by SK11 and SK12, other sites may also contribute to TTG1 phosphorylation since mutation of other sites slightly alters the overall phosphorylation patterns mediated by SK11 and SK12 (Fig. 4c, d). Thus, it is possible that SK11 and SK12 could modify multiple residues in TTG1 and fine-tune its functions in a subtle manner. Second, despite compromised binding of 4HA-TTG1 to the *GL2* locus in *ttg1-13 tt2-5 g4HA-TTG1* (Fig. 7b), 4HA-TTG1 binding to *GL2* is still detectable near the *GL2* transcription start site in this *tt2-5* background (Fig. 7b). This suggests that recruitment of TTG1 to the *GL2* locus is also dependent on other unknown regulator(s) in addition to TT2. As *tt2-5* exhibits relatively weak defects in seed characters than *ttg1-13* (Fig. 3a and Supplementary Fig. 10), it is possible that TT2 acts together with other co-factor(s) to recruit TTG1 at the *GL2* locus. Thus, it will be interesting to further investigate whether phosphorylation of TTG1 dynamically influences its interaction with multiple co-regulators to exert its specific activity in regulating seed characters.

## Methods

**Plant materials and growth conditions**. *Arabidopsis thaliana* plants were grown under long day conditions (16 h light/8 h dark) at 22 °C. The mutants *tt2-5* (SALK_005260), *tt8-4* (SALK_063334), *sk11* (SALK_014382), *sk12* (CS332559), and *gl2-3* (SALK_039825) are in the Columbia (Col-0) background. We obtained the Col-0 near-isogenic *ttg1-13* line by three backcrosses of the *ttg1-13* Rschew (RLD) line into Col-0 wild-type plants. All transgenic plants in the Col-0 background were generated through *Agrobacterium tumefaciens*-mediated transformation.

**Plasmid construction for plant transformation**. To construct *g4HA-TTG1*, the whole-genomic region of *TTG1* was amplified into two fragments (−2526 to −1; 1–2136), and inserted with a 4HA tag by overlapping PCR. *g4HA-TTG1^S215E* and *g4HA-TTG1^S215A* were then created based on the *g4HA-TTG1* fragment by overlapping PCR[46]. All these genomic fragments were cloned into the NT007 entry vector, and then introduced into the binary vector pBGW using the Gateway LR recombination reaction (Invitrogen). To construct *SK11:GUS* and *SK12:GUS*, the 2.6-kb and 2.3-kb 5′ upstream sequences of *SK11* and *SK12* were cloned into pBI101-GUS, respectively. To construct *SK11:SK11-GFP* and *SK11:SK11^E292K-GFP*, the coding sequences of the native *SK11* and *SK11^E292K* were cloned into pCAMBIA1302E that was early incorporated with the same *SK11* 5′ upstream sequence used for *SK11:GUS*, respectively. To construct *35S:TT2-GFP*, the coding sequence of *TT2* was cloned into pCAMBIA2302. The primers used for construction of plasmids for plant transformation were listed in Supplementary Table 1.

**Bikinin treatment**. For bikinin treatment, 25 µM bikinin (SML0094, Sigma) was dissolved in 1/2 MS pH 5.8 with 0.02% Silwet-77. To examine a long-term effect, bikinin solution was applied onto siliques at 4 days after pollination, and the resulting mature seeds were collected for further investigation. To examine an immediate effect of bikinin treatment in in vivo phosphorylation assay and gene expression analysis, siliques collected at 4 days after pollination were treated in bikinin or mock solution for 2 h.

**Expression analysis**. Total RNA from various tissues was extracted using FavorPrep Plant Total RNA Mini Kit (Favorgen) and reverse-transcribed using the M-MLV Reverse Transcriptase (Promega) according to the manufacturers' instructions. Quantitative real-time PCR was performed on three biological replicates using the CFX384 real-time PCR detection system with iQ SYBR Green Supermix (Bio-Rad). The expression of *U-BOX* was used as an internal control. Relative expression levels of genes were calculated by the ΔCt method[47]. We used

the reported primers[17] for examining the expression of fatty acid biosynthesis genes, while other primers are listed in Supplementary Table 1. GUS staining was carried out based on the published protocol[48] with minor modifications. Siliques were cut into 3 mm in length and briefly fixed in 90% acetone before being submerged in GUS staining solution under vacuum condition for 30 min. The samples were incubated in the staining solution for 8 h at 37 °C, after which they were cleared of chlorophyll, dehydrated, and placed in the clear solution (chloral hydrate/water/glycerol = 8 g/4 ml/2 ml) for microscopic observation.

**Microscopy and histochemical analysis**. Ruthenium red staining of seed coat mucilage was carried out as follows. Dry seeds were gently washed once with water, and the rehydrated seeds were stained in 0.01% (w/v) ruthenium red solution at room temperature for 1 h for subsequent microscopic observation. For DMACA staining, dry seeds were stained with the DMACA reagent (2% (w/v) DMACA in 3 M HCl and 50% (w/v) methanol) at room temperature under dark conditions for 12 h, and then washed three times with 70% (v/v) ethanol. Treated seeds were observed using Nikon SMZ1500 (Nikon) or Axio Imager 2 (Carl Zeiss).

**Pull-down assay**. For LC-MS/MS analysis, Col-0 wild-type siliques were collected 4 days after pollination for extraction of total protein. The total protein extract was incubated with the recombinant MBP-TTG1 immobilized onto amylose resin for 3 h at 4 °C. After washing, the pull-downed protein mixture was eluted and analyzed by LC-MS/MS using a TripleTOF 5600 System (AB Sciex). For in vitro pull-down assay, the complementary DNAs (cDNAs) encoding TT2, TTG1, and TTG1 mutated forms were cloned into pMAL-c2x vector to generate MBP-tagged proteins. The cDNAs encoding SK11 and SK12 were cloned into pGEX-4T-1 to generate GST-tagged proteins, while the cDNA encoding TT2 was cloned into pET28a to generate the His-tagged protein.

The primers used for these constructs were listed in Supplementary Table 1. All proteins with various tags were expressed by inducing transformed *Escherichia coli* stain Rossetta (DE3) with isopropyl β-D-1-thiogalactopyranoside (IPTG) at 25 °C for 5 h. The soluble fusion proteins were extracted and immobilized on glutathione sepharose beads (Amersham Biosciences), amylose beads (NEB), or TALON metal affinity resin (Clontech). Proteins retained on the beads were subsequently resolved by SDS-polyacrylamide gel electrophoresis and detected with various antibodies (anti-GFP: ab290, Abcam, 1:2000 dilution; anti-MBP-HRP: E8038S, NEB, 1:10,000 dilution; anti-His: sc-8036 HRP, Santa Cruz, 1:3000 dilution).

**Co-immunoprecipitation assay**. Siliques 4 days after pollination in various genetic backgrounds were collected, and total proteins were extracted by extraction buffer (50 mM Tris-HCl pH 7.5, 150 mM NaCl, 1 mM EDTA, 5% glycerol, 0.5% Triton X-100, 1 mM PMSF, proteinase inhibitor cocktail, and 25 μM MG132). Protein extracts were incubated with anti-HA agarose beads (A2095, Sigma) at 4 °C for 2 h, and then washed four times by extraction buffer. Immunoprecipitated proteins and total protein extracts as inputs were resolved by SDS-polyacrylamide gel electrophoresis and detected by anti-GFP antibody (anti-GFP: ab290, Abcam, 1:2000 dilution) and anti-HA HRP (anti-HA: sc-7392 HRP, Santa Cruz, 1:2000 dilution).

**Yeast two-hybrid assay**. The coding sequences of *SK11*, *SK12*, *TTG1*, *TTG1^{S215A}*, and *TTG1^{S215E}* were amplified and cloned into pGADT7 (Clontech), while the coding sequences of *TT2*, *TTG1*, *TTG1^{S215A}*, and *TTG1^{S215E}* were amplified and cloned into pGBKT7 (Clontech). All primers used for these constructs were listed in Supplementary Table 1. Yeast two-hybrid assays were performed using the Yeastmaker Yeast Transformation System 2 (Clontech).

**BiFC assay**. The coding sequences of *TTG1*, *TTG1^{S215A}*, and *TTG1^{S215E}* were fused to cYPF in pXY104, while those of *TT2*, *SK11*, and *SK12* were fused to nYFP in pXY106. The primers for these constructs were listed in Supplementary Table 1. The plasmids were transformed into *A. tumefaciens*, which was then infiltrated into *Nicotiana benthamiana* leaves. Leaves were observed 2 days after infiltration under a confocal microscope.

**In vitro kinase assay**. Kinases and substrates were expressed by inducing transformed *E. coli* stain Rossetta (DE3) with IPTG. After purification, they were incubated in reaction buffer (50 mM Tris-HCl pH 7.5, 50 mM NaCl, 12 mM MgCl$_2$, 1 mM DTT) with or without 1 mM ATP for 1 h at 37 °C. The reaction products were resolved by SDS-polyacrylamide gel electrophoresis with or without 50 μM Phos-tag (Wako Pure Chemical Industries).

**Measurement of fatty acids**. Fatty acids in seeds were measured by gas chromatography as follows. Acid-catalyzed transmethylation reaction on 10 mg of intact seeds was carried out in 5% H$_2$SO$_4$ in methanol (v/v) at 90 °C for 3 h, while 10 μg of heptadecanoic acid was used as an internal control. Fatty acid methyl ester was extracted by hexane and analyzed through a GC-2010 GCMS system (Shimadzu, Japan) equipped with a HP-88 fused silica column (Agilent, USA). The pulsed split mode (250 °C) and helium gas (flow rate of 1 ml/min) were used as the injector condition and carrier gas, respectively, while the pulsed pressure was set at 15 psi for 0.5 min. Scan range was *m/z* 40–500. The oven program was: 150 °C for 3 min, ramp of 7 °C per min to 240 °C, and keep for 5 min. ChemStation software (Shimadzu, Japan) was used for data acquisition and processing, and compounds were identified by comparison of mass spectrum with authentic standards and the NIST/EPA/NIH mass spectral library v2.0.

**ChIP assay**. ChIP assay was carried out on siliques in various genetic backgrounds 4 days after pollination following the reported protocol[49] with minor modifications. The extracted chromatin was sonicated to produce DNA fragments between 200 and 500 bp. The solubilized chromatin was incubated with anti-HA agarose beads (A2095, Sigma) for 2 h at 4 °C. A genomic fragment of *TUBLIN2 (TUB2)* was amplified as a control. Chromatin immunoprecipitation assays were repeated with three biological replicates. The primers designed for ChIP-PCR were listed in Supplementary Table 1.

**EMSA assay**. Electrophoretic mobility shift assay was performed using LightShift Chemiluminescent EMSA kit (Pierce). Recombinant proteins were produced as described in the section of "pull-down assay." The native probe was produced by PCR using 5′-biotinylated primers, while the mutated probe was produced by site-directed mutagenesis via overlapping PCR. Competitors were produced by PCR using unlabeled primers. The primers for synthesizing these probes were listed in Supplementary Table 1.

**Data availability**. The MS proteomics data have been deposited to the ProteomeXchange Consortium via the PRIDE partner repository with the data set identifier PXD008609. The authors declare that all other data supporting the findings of this study are available within the article and its supplementary information files or from the corresponding author upon reasonable request.

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

## Acknowledgements

We thank the Arabidopsis Biological Resource Centre for providing seeds. We thank Dr Liang Zhe for sequence analysis, the Protein and Proteomics Centre (PPC) in the Department of Biological Sciences, National University of Singapore for mass spectrometry service, and members of the Yu lab for discussion and comments on the manuscript. This work was supported by Academic Research Fund (MOE2015-T2-2-002) from the Ministry of Education-Singapore, the Singapore National Research Foundation Investigatorship Programme (NRF-NRFI2016-02), and the intramural research support from National University of Singapore and Temasek Life Sciences Laboratory.

## Author contributions

C.L. and H.Y. conceived and designed the study. C.L., B.Z., B.C., and L.J. performed the experiments. C.L. and H.Y. analyzed the data and wrote the paper.

## Additional information

**Competing interests:** The authors declare no competing financial interests.

