## [Peer Review File · Nature Communications]

Reviewers' comments:

Reviewer #1 (Remarks to the Author):

Li et al have produced a strong manuscript showing that the phosphorylation of TTG1 by the SHAGGY KINASES 11 and 12 affects its interaction with TT2, and by this means affects carbon partitioning within the developing seed.

The data in the manuscript are compelling, with strong biochemical analysis being supported by phenotypic observations, gene expression analysis, metabolite analysis and genetic analysis, all of which converge to support the conclusions drawn by the authors. In addition the manuscript is carefully written and pleasant to read, and is well illustrated.

Specific points pertaining to results:

General point:

The authors use many transgenic lines, however, it worries me that data are only presented for one line per construct. This appears a little dangerous, and other lines should strictly be presented. Maybe this could be done in supplementary data to limit the size of the main figures (which are already very large)

Figure 1

- b) Why is there BiFC signal in the cytoplasm when TTG1 should be localised to the nucleus ? The authors should maybe comment on this
- d) I don't understand why the HA panel appears to be a composite of two gels. This makes me a little uncomfortable. Could the authors comment?

Figure 3

- a) For the SK11 :SK11E292K transgenic, it is not clear whether this is in a wild-type or a double mutant background, and in any case, both should really be shown. There is a clear difference in mucilage structure in this line (it looks a bit like the tt2-5 line), which should be mentioned.
- Bikinin treatments (also relevant to figure 4b). It's amazing that just painting bikinin onto siliques actually works (although I have to say that the results really do appear to be quite convincing). What is known about bikinin stability and mobility? Can any assay be done to verify that the effects seen are really due to bikinin in seeds?

Figure 5

- There is slight mucilage secretion and anthocyanin biosynthesis associated with the "rescue" of ttg1-13 with the TTG1S215E protein version. This should be discussed.
- Another important point is that in theory, if the hypothesis of the authors is correct, this construction should rescue the phenotype of the sk11/sk12 double mutant. This would be a very nice genetic proof of the fact that TTG1 acts downstream of SK11 and SK12 which would help with the fact that this is only currently supported by Bikinin treatments (see above). It would presumably not be too difficult to test?

Figure 6

- b) Why would TTG1S215A be more efficiently pulled down by TT2 than wild-type TTG1 in vitro? The authors need to explain this because it suggests that this mutation could be "forcing" an interaction with TT2 and that the phenotypes in plants expressing this variant may not therefore be very easy to

interpret simply in terms of phosphorylation/de-phosphorylation of TTG1.

-Are the phenotypic effects of expressing TTG1S215A completely dependent upon the presence of TT2? Again this would be easy to test in a *ttg2* mutant background and this would strengthen the conclusions of the paper

-d) Why is TTG1S215A not included in this experiment?

Figure 7

g) The fact that bikinin treatment causes a stronger increase in GL2 expression than the double mutant *sk11/sk12* suggests that there may be redundancy with other SKs. This should be discussed.

Points relating to the manuscript:

Major points

Globally I think that the conclusions of this work are clear and are supported by the data. However I have a problem with the discussion and impact of the results.

Firstly, although TTG1, SK11 and SK12 are expressed throughout the seed, TT2 is basically seed coat specific, suggesting that the increase in oil content seen when TT2 and TTG1 function is lost is likely due to a "default" redirection/remobilisation of carbon to the embryo in the absence of normal seed coat differentiation (to test whether this is indeed the case the authors should ideally investigate whether the lipid accumulation defects seen in their various mutants and lines are dependent upon maternal or zygotic phenotypes). I think this needs to be pointed out clearly somewhere in the discussion. Seed coat development is important for seed survival, so is seed coat differentiation vs embryo really a viable point of control for carbon flux? The results presented here seem to suggest that it could be, but again this is not discussed.

Secondly, I have no problem in believing that phosphorylation of TTG1 by SK11 and SK12 somehow regulates the formation of the TTG1/TT2 complex and that this alters the degree of differentiation of the seed coat and thus, by default, carbon flux to the embryo. However the real question is why? How is this controlled? Is the activity of SK11 and SK12 altered by the nutrient status of the plant or by other environmental factors? It seems to me that without some information (or at the very least some discussion) of this aspect, the results in this paper simply serve to push back by one level the control of seed coat differentiation, without adding insight into its metabolic/environmental control. This is a shame.

Thirdly, in the discussion no reference is made to other SK targets which could be affected by the mutations/manipulations carried out in this work. SK11 and SK12 have other known targets (and indeed developmental functions) and these should be mentioned not only as a courtesy to the researchers who have uncovered them, but also because ultimately they could impact some of the phenotypes observed.

More minor points:

First paragraph of introduction. At seed maturity the seed coat is 100% dead. Maternal tissues cannot therefore perceive environmental signals. The endosperm can (it is alive). The last sentence of this paragraph should therefore be rephrased.

Page 4, fourth line from the bottom. I think the authors want to say "have so far not yet been obtained"

Page 17, eighth line from bottom, replace "irrelevant" with "insensitive"

There are one or two places in the introduction where "seat" is written instead of "seed". These should be changed.

Reviewer #2 (Remarks to the Author):

What are the major claims of the paper?

The manuscript describes studies of the role of TTG1 in seed development. Evidence is provided that TTG1 is phosphorylated by SK11 and SK12 and that unphosphorylated TTG1 interacts with TT2 to regulate GL2 expression whereas phosphophorylation of TTG1 disrupts this interaction. It is proposed that phosphorylation of TTG1 causes a reduction in mucilage and flavonoid pigment production in the seed coat and enhanced fatty acid biosynthesis in the embryo and, therefore, that SK11/12 controls carbon partitioning within the seed.

Are they novel and will they be of interest to others in the community and the wider field?

The biological questions addressed by the manuscript are important. Understanding how assimilates are partitioned in the seed has implications for the control of crop yield. Thus, the manuscript may be of interest to a broad audience of plant biologists. The manuscript also presents another example of a transcription factor whose function is altered by its phosphorylation which may be of interest to those studying transcriptional regulation.

Is the work convincing, and if not, what further evidence would be required to strengthen the conclusions?

A strength of the manuscript is that several of the conclusions are very well supported. Multiple approaches are often used to provide evidence in support of a conclusion. For example, in vitro pull-down experiments, in vivo coimmunoprecipitation experiments, yeast two hybrid assays, and bimolecular fluorescence complementation assays are used to demonstrate physical interactions between proteins.

One weaknesses of the manuscript is that it does not show where in the seed the interactions between SK11/12 and TTG1 occur to control fatty acid biosynthesis. Given that TT2 mRNA appears to accumulate specifically in the seed coat, it is likely that the interactions between unphosphorylated TTG1 and TT2 occurs in the seed coat to regulate mucilage and flavonoid pigment synthesis. It may also be that SK11/12 phosphorylates TTG1 in the seed coat to inhibit mucilage and flavonoid pigment synthesis, although this is not shown directly. It is not clear, however, where phosphorylated TTG1 acts to affect fatty acid biosynthesis. The authors describe the phosphorylation of TTG1 by SK11/12 as a molecular switch that causes TTG1 to change its function from promoting mucilage and flavonoid pigment synthesis to fatty acid biosynthesis. The term molecular switch implies that phosphorylation of TTG1 in seed coat cells causes assimilates used for mucilage synthesis to be shunted towards the embryo to enhance fatty acid biosynthesis. However, SK11/12 and TTG1 mRNAs accumulate in seed coat, embryo, and endosperm regions of the seed. Therefore, it is possible that SK11/12 and TTG1 work directly in the embryo to control fatty acid biosynthesis by creating a sink for assimilates. Reciprocal crosses between sk11 sk12 and wild type might help to resolve this issue. Overall, spatial aspects of the model shown in Fig. 8 are not adequately discussed in the manuscript. Similarly, the ChIP experiments summarized in Fig. 7 are done with siliques which consists of embryo, endosperm, seed coat, and fruit tissues, and it is not stated if TTG1 or TT2 are expressed in fruit tissues, such as the value. Therefore, the binding patterns for TTG1 and TT2 may represent a mosaic of binding that occurs in embryo, endosperm, seed coat, and fruit tissues, and the GL2 gene may be regulated differently by TTG1 in different tissues.

Other points that should be addressed follow.

1. The data in Fig. 3 indicate that TT2 acts downstream of SK11/12, but they do not show genetically that TTG1 acts downstream of SK11/12.
2. I do not understand what the gradient symbols indicate for the AD proteins in the yeast two hybrid experiments shown in Figs. 1 and 6. Are different promoters used to drive the AD constructs so that a gradient of AD protein levels are produced?
3. Supplementary Fig. 6 is numbered incorrectly.

On a more subjective note, do you feel that the paper will influence thinking in the field?

I believe the manuscript would advance the field significantly if it can be shown that SK11/12 and TTG1 work solely in the seed coat as a molecular switch to influence carbon partitioning in the seed.

Reviewer #3 (Remarks to the Author):

This paper builds an interesting gene regulatory model concerning regulation of major biosynthetic pathways in seeds by transcription factors. It would be of broad interest and use to plant scientists. The model is mostly well supported by the novel data presented. The experiments appear to be thorough and well designed.

Major comments

- I am convinced by the data that the proposed role of SK11 in the overall model (Fig 8) is justified. However, the role of SK12 is not as clear because of the differing band patterns within the phostag gels which indicate that SK12 phosphorylates different residues than SK11 during in vitro incubation with TTG1 in kinase assays. This is evident in fig 4a and 4d. Also the loading in 4d does not equal for the S215A lane which may have made it seem like there is a difference there.
- The phostag immunoblot in Fig 4b lacks controls which would enable an important comparison with 4a. Is the band pattern obtained from in vivo phosphorylation similar to that obtained during in vitro kinase assays with SK11 or SK12? The blot should contain samples similar to lanes 1, 3 and 4 in 4a.
- Fig 2b should include data from ttg1 samples and ideally tt2?

Minor comments

Introduction:

- "...specifically mediates TTG1 effect on controlling carbon flow". Other possible effects of this phosphorylation even are possible and not tested. Omit specifically.
- The term "master switch" seems unhelpful. What is a master switch vs a switch? Replace with control mechanism or something more precise.

Results:

- "...redundant biological functions" there is no evidence for this. Change to overlapping.
- Can the developmental pattern of SK11 and TTG1 gene expression be interpreted with regards to the model? Is there a developmental change in carbon distribution between fatty acids and maternal tissue?
- Can the statistical analysis of figure 2b and g be done with ANOVA and post-hoc testing in order to compare all samples (e.g. indicated with different letters)
- "...compromised TTG1 binding to the GL2 locus" Change compromise to weakened.
- "This expression trend generally followed that of TTG1 and TT2" The patterns between GL2 and TTG1 is not similar.

-Stats should be applied to Fig 7g.

Discussion:

-The mechanism of TTG1 is referred to several times as carbon partitioning between metabolic pathways. Is it not more likely to represent carbon partitioning between tissues (maternal vs embryo)? For example what about seeds which don't store much oil?

-Is Ser215 of TTG1 conserved among plants? The adjacent proline may also be involved in directing phosphorylation.

Response to Reviewers

We would like to thank the three reviewers for the time committed in reviewing this manuscript, and for all the detailed suggestions on improving this manuscript. We have substantially revised the manuscript to fully address the reviewers' concerns and criticisms as follows.

Reviewer 1

Li et al have produced a strong manuscript showing that the phosphorylation of TTG1 by the SHAGGY KINASES 11 and 12 affects its interaction with TT2, and by this means affects carbon partitioning within the developing seed.

The data in the manuscript are compelling, with strong biochemical analysis being supported by phenotypic observations, gene expression analysis, metabolite analysis and genetic analysis, all of which converge to support the conclusions drawn by the authors. In addition the manuscript is carefully written and pleasant to read, and is well illustrated.

>Reviewer:

(General point) The authors use many transgenic lines, however, it worries me that data are only presented for one line per construct. This appears a little dangerous, and other lines should strictly be presented. Maybe this could be done in supplementary data to limit the size of the main figures (which are already very large).

>Authors:

We obtained and characterized many independent lines for each construct. As suggested by this reviewer, we have presented the phenotypes of other representative transgenic lines that harbored the key constructs (e.g. *g4HA-TTG1*, *g4HA-TTG1^{S215E}*, *g4HA-TTG1^{S215A}*, *SK11:SK11-GFP* and *SK11:SK11^{E292K}-GFP*) with possibly only one T-DNA insertion site based on the segregation ratio in the revised Supplementary Figs. 4, 5, and 12.

>Reviewer:

(Figure 1b)-Why is there BiFC signal in the cytoplasm when TTG1 should be localised to the nucleus? The authors should maybe comment on this.

>Authors:

Our experiments have repeatedly shown the BiFC signal of SK11/12-TTG1 interaction in both the cytoplasm and nucleus. In addition, we have similarly observed localization of TTG1-mCherry in both the cytoplasm and nucleus of *N. benthamiana* leaf epidermal cells (new Supplementary Fig. 3a). More importantly, immunoblot analysis of proteins extracted from *ttg1-13 g4HA-TTG1 Arabidopsis* siliques has confirmed TTG1 localization in both cytosolic and nuclear fractions (new Supplementary Fig. 3b). These results support that TTG1 is localized in both the cytoplasm and nucleus.

>Reviewer:

(Figure 1d)-I don't understand why the HA panel appears to be a composite of two gels. This makes me a little uncomfortable. Could the authors comment?

>Authors:

This HA panel was originated from the same gel blot. The vertical line was incidentally produced by the X-Ray Film Processor. We have included the original uncropped image (Fig. 1d) in Supplementary Fig. 18.

>Reviewer:

(Figure 3a)-For the SK11:SK11E292K transgenic, it is not clear whether this is in a wild-type or a double mutant background, and in any case, both should really be shown. There is a clear difference in mucilage structure in this line (it looks a bit like the *tt2-5* line), which should be mentioned.

>Authors:

In Fig 3a, *SK11:SK11^{E292K}-GFP* indicates the transgenic line created in the wild-type background. We have also obtained *sk11 sk12 SK11:SK11^{E292K}-GFP* by crossing *SK11:SK11^{E292K}-GFP* with *sk11 sk12*. As expected, the gain-of-function *SK11:SK11^{E292K}-GFP* line suppressed *sk11 sk12* seed phenotypes (Fig. 3b and new Supplementary Fig. 5c-bottom panel). As suggested by this reviewer, we have also mentioned the similar mucilage structure between *tt2-5* and *SK11:SK11^{E292K}-GFP* in the text.

>Reviewer:

(Figure 3)-Bikinin treatments (also relevant to figure 4b). It's amazing that just painting bikinin onto siliques actually works (although I have to say that the results really do appear to be quite convincing). What is known about bikinin stability and mobility? Can any assay be done to verify that the effects seen are really due to bikinin in seeds?

>Authors:

Although bikinin stability and mobility have yet been investigated in detail, bikinin treatment of plant tissues with various durations has been successfully applied in several recent studies (Kondo et al., 2015; Guo et al., 2017; Zhu et al., 2017), indicating that this small molecule is able to penetrate into plant cells. In this study, we have carefully compared the effects of bikinin and mock treatment of siliques on developing seeds. The difference in seed phenotypes shown in Fig. 3d should be attributable to bikinin because the only different component in bikinin and mock solutions was 25 uM bikinin as described in Methods. Consistently, we have found that phosphorylation of 4HA-TTG1 was inhibited in protein extracts from bikinin-treated siliques (Fig. 4b), indicating that bikinin treatment of siliques affects TTG1 phosphorylation in seeds

Relevant references:

Kondo et al. (2015) A novel system for xylem cell differentiation in *Arabidopsis thaliana*. *Mol. Plant* 8, 612–621.

Guo et al. (2017) Identification of cyst nematode B-type CLE peptides and modulation of the vascular stem cell pathway for feeding cell formation. *PLoS Pathog.* 13, e1006142.

Zhu et al. (2017) The F-box protein KIB1 mediates brassinosteroid-induced inactivation and degradation of GSK3-like kinases in *Arabidopsis*. *Mol. Cell* 66, 648–657.

>Reviewer:

(Figure 5)-There is slight mucilage secretion and anthocyanin biosynthesis associated with the “rescue” of *ttg1-13* with the TTG1S215E protein version. This should be discussed.

>Authors:

As suggested by this reviewer, we have described these phenotypes and provided a relevant discussion in the revised manuscript.

>Reviewer:

(Figure 5)-Another important point is that in theory, if the hypothesis of the authors is correct, this construction (TTG1^{S215E}) should rescue the phenotype of the *sk11/sk12* double mutant. This would be a very nice genetic proof of the fact that TTG1 acts downstream of SK11 and SK12 which would help with the fact that this is only currently supported by BIKININ treatments (see above). It would presumably not be too difficult to test?

>Authors:

The effect of the phosphorylation-mimicking form TTG1^{S215E} can only be tested in *ttg1-13* mutants, where the wild-type TTG1 protein is not functional. We agree with this reviewer that it would be nicer to examine whether *ttg1-13 g4HA-TTG1^{S215E}* rescues the phenotypes of *sk11 sk12*. However, because *TTG1* (*AT5g24520*) and *SK11* (*At5g26751*) are very close loci in chromosome 5, we were unable to obtain the genetic material (*sk11 sk12 ttg1-13 g4HA-TTG1^{S215E}*). This is actually the primary reason why we applied BIKININ treatment to test the genetic interaction between *TTG1* and *SK11/SK12* as described in the text.

>Reviewer:

(Figure 6b)-Why would TTG1S215A be more efficiently pulled down by TT2 than wild-type TTG1 in vitro? The authors need to explain this because it suggests that this mutation could be “forcing” an interaction with TT2 and that the phenotypes in plants expressing this variant may not therefore be very easy to interpret simply in terms of phosphorylation/de-phosphorylation of TTG1.

>Authors:

Our in vitro pull-down assays (Fig. 6b) demonstrated that TT2-His had a stronger interaction with MBP-TTG1^{S215A} than MBP-TTG1. To test the in vivo effect of S215A, we have performed Co-IP analysis and found that TT2-GFP indeed interacted more strongly with 4HA-TTG1^{S215A} than 4HA-TTG1 (new Fig. 6d). It is possible that in addition to the effect of dephosphorylation, mutation of Ser(S)215 to Ala (A) may also affect other biochemical characteristics that influence the interaction between TTG1 and TT2. Nevertheless, this observation does not affect the main conclusion that phosphorylation of TTG1 at Ser 215 abolishes the interaction between TTG1 and TT2 in siliques, which is supported by all Fig. 6a-d results.

>Reviewer:

(Figure 6b)-Are the phenotypic effects of expressing TTG1S215A completely dependent upon the presence of TT2? Again this would be easy to test in a *tt2* mutant background and this would strengthen the conclusions of the paper

>Authors:

As suggested by this reviewer, we have examined the seed phenotypes of *tt2-5 ttg1-13 g4HA-TTG1^{S215A}*, and found that *tt2-5 ttg1-13 g4HA-TTG1^{S215A}* exhibited the identical phenotypes of seed color and fatty acid to *tt2-5* (new Supplementary

Fig. 15), substantiating that TTG1 interaction with TT2 is important for mediating TTG1 effect on seed phenotypes.

>Reviewer:

(Figure 6d)-Why is TTG1S215A not included in this experiment?

>Authors:

We have repeated CoIP assays, and included the result from g4HA-TTG1^{S215A} in the revised manuscript (new Fig. 6d).

>Reviewer:

(Figure 7g)-The fact that bikinin treatment causes a stronger increase in GL2 expression than the double mutant *sk11/sk12* suggests that there may be redundancy with other SKs. This should be discussed.

>Authors:

We agree with this reviewer, and have provided the relevant discussion in the text. As bikinin specifically inhibits subgroups I and II GSK3-like kinases, including SK11, SK12, and other 4 homologs in *Arabidopsis*, stronger expression of *GL2* in bikinin-treated samples than *sk11 sk12* could be partially due to bikinin effects on other SKs, which may share a similar role of SK11 and SK12 in affecting *GL2* expression.

>Reviewer:

Major point

Globally I think that the conclusions of this work are clear and are supported by the data. However I have a problem with the discussion and impact of the results.

Firstly, although TTG1, SK11 and SK12 are expressed throughout the seed, TT2 is basically seed coat specific, suggesting that the increase in oil content seen when TT2 and TTG1 function is lost is likely due to a “default” redirection/remobilisation of carbon to the embryo in the absence of normal seed coat differentiation (to test whether this is indeed the case the authors should ideally investigate whether the lipid accumulation defects seen in their various mutants and lines are dependent upon maternal or zygotic phenotypes). I think this needs to be pointed out clearly somewhere in the discussion. Seed coat development is important for seed survival, so is seed coat differentiation vs embryo really a viable point of control for carbon flux? The results presented here seem to suggest that it could be, but again this is not discussed.

>Authors:

We agree that this reviewer raised important issues for us to improve the relevant discussions. As suggested by this reviewer, we have performed reciprocal genetic crosses between WT and *sk11 sk12*, and found that SK11 and SK12 regulate fatty acid levels maternally (new Supplementary Fig. 17). This result, together with the specific function of TT2 in seed coat and the maternal effects of TTG1 and GL2 on fatty acid levels in seeds (Chen et al., 2015; Shi et al., 2012), suggests that SK11/12-mediated TTG1 phosphorylation maternally regulates carbon partitioning in seeds. We have incorporated the new results and relevant discussions in the revised manuscript.

Although our results suggest that site-specific phosphorylation of TTG1 by SK11/SK12 is essential for regulating carbon partitioning in seeds, all loss-of-function mutants discussed in this study still produce viable seeds,

indicating that the defect in carbon flux might not directly affect seed viability under normal growth conditions. This has been discussed in the revised manuscript. We agree that the effect of carbon flux on seed survival is certainly an interesting topic, and will further investigate it under various environmental conditions in our future studies.

Relevant references:

Chen et al. (2015) TRANSPARENT TESTA GLABRA 1 regulates the accumulation of seed storage reserves in Arabidopsis. Plant Physiol. 169, 391-402.

Shi et al. (2012). Arabidopsis *glabra2* mutant seeds deficient in mucilage biosynthesis produce more oil. Plant J. 69, 37-46.

>Reviewer:

Secondly, I have no problem in believing that phosphorylation of TTG1 by SK11 and SK12 somehow regulates the formation of the TTG1/TT2 complex and that this alters the degree of differentiation of the seed coat and thus, by default, carbon flux to the embryo. However the real question is why? How is this controlled? Is the activity of SK11 and SK12 altered by the nutrient status of the plant or by other environmental factors? It seems to me that without some information (or at the very least some discussion) of this aspect, the results in this paper simply serve to push back by one level the control of seed coat differentiation, without adding insight into its metabolic/environmental control. This is a shame.

>Authors:

As suggested by this reviewer, we have included the relevant discussions in the revised manuscript. By now there have been very few studies of the signals upstream of SK11 and SK12. It has been reported that SK11 (also known as ASK α) is induced by abiotic stresses, such as salt stress, to regulate stress tolerance by activating glucose-6-phosphate dehydrogenase (G6PD) and affecting reactive oxygen species (ROS) levels (Dal Santo et al., 2012). In addition, another study has shown that SK11 phosphorylates G6PD, thus constituting an immune signaling module in response to pathogenic microbes and linking protein phosphorylation cascades to metabolic regulation (Stampfl et al., 2016). It's noteworthy that G6PD catalyzes a key step of the oxidative pentose phosphate pathway (OPPP) that generates NADPH necessary for fatty acid biosynthesis. We have included the discussion on our findings in the context of these previous literatures.

Relevant references:

Dal Santo et al. (2012) Stress-induced GSK3 regulates the redox stress response by phosphorylating glucose-6-phosphate dehydrogenase in Arabidopsis. Plant Cell 24, 3380-3392.

Stampfl et al. (2016) The GSK3/Shaggy-like kinase ASK α contributes to pattern-triggered immunity. Plant Physiol. 171, 1366-1377.

>Reviewer:

Thirdly, in the discussion no reference is made to other SK targets which could be affected by the mutations/manipulations carried out in this work. SK11 and SK12

have other known targets (and indeed developmental functions) and these should be mentioned not only as a courtesy to the researchers who have uncovered them, but also because ultimately they could impact some of the phenotypes observed.

>Authors:

To our knowledge, among all class I GSK3s, only one in vivo target of SK11, G6PD, has so far been reported (Dal Santo et al., 2012). As suggested by this reviewer, we have included the discussion on this target in the revised manuscript.

>Reviewer:

More minor points:

First paragraph of introduction. At seed maturity the seed coat is 100% dead. Maternal tissues cannot therefore perceive environmental signals. The endosperm can (it is alive). The last sentence of this paragraph should therefore be rephrased.

>Authors:

As suggested by this reviewer, we have rephrased the sentence.

>Reviewer:

Page 4, fourth line from the bottom. I think the authors want to say “have so far not yet been obtained”

>Authors:

We thank this reviewer for the correction, and have revised the sentence.

>Reviewer:

Page 17, eighth line from bottom, replace “irrelevant” with “insensitive”. There are one or two places in the introduction where “seat” is written instead of “seed”. These should be changed.

>Authors:

We thank this reviewer for these corrections, and have revised them accordingly.

Reviewer 2

What are the major claims of the paper?

The manuscript describes studies of the role of TTG1 in seed development. Evidence is provided that TTG1 is phosphorylated by SK11 and SK12 and that unphosphorylated TTG1 interacts with TT2 to regulate GL2 expression whereas phosphorylation of TTG1 disrupts this interaction. It is proposed that phosphorylation of TTG1 causes a reduction in mucilage and flavonoid pigment production in the seed coat and enhanced fatty acid biosynthesis in the embryo and, therefore, that SK11/12 controls carbon partitioning within the seed.

Are they novel and will they be of interest to others in the community and the wider field?

The biological questions addressed by the manuscript are important. Understanding how assimilates are partitioned in the seed has implications for the control of crop yield. Thus, the manuscript may be of interest to a broad audience of plant biologists. The manuscript also presents another example of a transcription factor whose function

is altered by its phosphorylation which may be of interest to those studying transcriptional regulation.

Is the work convincing, and if not, what further evidence would be required to strengthen the conclusions?

A strength of the manuscript is that several of the conclusions are very well supported. Multiple approaches are often used to provide evidence in support of a conclusion. For example, *in vitro* pull-down experiments, *in vivo* coimmunoprecipitation experiments, yeast two hybrid assays, and bimolecular fluorescence complementation assays are used to demonstrate physical interactions between proteins.

>Reviewer:

One weaknesses of the manuscript is that it does not show where in the seed the interactions between SK11/12 and TTG1 occur to control fatty acid biosynthesis. Given that TT2 mRNA appears to accumulate specifically in the seed coat, it is likely that the interactions between unphosphorylated TTG1 and TT2 occurs in the seed coat to regulate mucilage and flavonoid pigment synthesis. It may also be that SK11/12 phosphorylates TTG1 in the seed coat to inhibit mucilage and flavonoid pigment synthesis, although this is not shown directly. It is not clear, however, where phosphorylated TTG1 acts to affect fatty acid biosynthesis. The authors describe the phosphorylation of TTG1 by SK11/12 as a molecular switch that causes TTG1 to change its function from promoting mucilage and flavonoid pigment synthesis to fatty acid biosynthesis. The term molecular switch implies that phosphorylation of TTG1 in seed coat cells causes assimilates used for mucilage synthesis to be shunted towards the embryo to enhance fatty acid biosynthesis. However, SK11/12 and TTG1 mRNAs accumulate in seed coat, embryo, and endosperm regions of the seed. Therefore, it is possible that SK11/12 and TTG1 work directly in the embryo to control fatty acid biosynthesis by creating a sink for assimilates. Reciprocal crosses between *sk11 sk12* and wild type might help to resolve this issue. Overall, spatial aspects of the model shown in Fig. 8 are not adequately discussed in the manuscript. Similarly, the ChIP experiments summarized in Fig. 7 are done with siliques which consists of embryo, endosperm, seed coat, and fruit tissues, and it is not stated if TTG1 or TT2 are expressed in fruit tissues, such as the value. Therefore, the binding patterns for TTG1 and TT2 may represent a mosaic of binding that occurs in embryo, endosperm, seed coat, and fruit tissues, and the GL2 gene may be regulated differently by TTG1 in different tissues.

>Authors:

We agree that this reviewer raised valid points for us to improve the manuscript. As suggested by this reviewer, we have performed reciprocal genetic crosses between WT and *sk11 sk12*, and found that SK11 and SK12 maternally regulate fatty acid levels (new Supplementary Fig. 17). In addition, previous studies have shown the maternal effects of TTG1 and GL2 on fatty acid levels in seeds (Chen et al., 2015; Shi et al., 2012), and that TT2 is predominantly expressed in the seed coat and controls seed coat development (Nesi et al., 2001; Gonzalez et al., 2009). Thus, although SK11/12 and TTG1 mRNAs are expressed in whole seeds, our results, together with previous findings, support that SK11/12-mediated TTG1 phosphorylation and its interaction with TT2 as well as their effects on GL2 maternally regulate carbon partitioning in seeds. We have incorporated the new results and relevant discussions in the revised manuscript.

Relevant references:

Chen et al. (2015) **TRANSPARENT TESTA GLABRA 1 regulates the accumulation of seed storage reserves in Arabidopsis. Plant Physiol. 169, 391-402.**

Shi et al. (2012). **Arabidopsis *glabra2* mutant seeds deficient in mucilage biosynthesis produce more oil. Plant J. 69, 37-46.**

Nesi et al. (2001) **The Arabidopsis TT2 gene encodes an R2R3 MYB domain protein that acts as a key determinant for proanthocyanidin accumulation in developing seed. Plant Cell 13, 2099-2114.**

Gonzalez et al. (2009) **TTG1 complex MYBs, MYB5 and TT2, control outer seed coat differentiation. Dev. Biol. 325, 412-421.**

>Reviewer:

Other points that should be addressed follow.

1. The data in Fig. 3 indicate that TT2 acts downstream of SK11/12, but they do not show genetically that TTG1 acts downstream of SK11/12.

>Authors:

Because *TTG1* (AT5g24520) and *SK11* (At5g26751) are very close loci in chromosome 5, we were unable to create the relevant genetic material (*sk11 sk12 ttg1-13*). This is the primary reason why we applied bikinin treatment to test the genetic interaction between *TTG1* and *SK11/SK12* as described in Fig. 3. Bikinin treatment, which inhibits SK11/SK12, almost had no effect on seed coat color and fatty acid levels of *ttg1-13* (Fig. 3e,g), indicating that *TTG1* acts downstream of *SK11* and *SK12*.

>Reviewer:

2. I do not understand what the gradient symbols indicate for the AD proteins in the yeast two hybrid experiments shown in Figs. 1 and 6. Are different promoters used to drive the AD constructs so that a gradient of AD protein levels are produced?

>Authors:

The gradient symbols represent the ten-fold serial dilutions of transformed yeast cells as indicated in the corresponding figure legends.

>Reviewer:

3. Supplementary Fig. 6 is numbered incorrectly.

>Authors:

We thank this reviewer for this correction, and have revised it accordingly.

>Reviewer:

On a more subjective note, do you feel that the paper will influence thinking in the field?

I believe the manuscript would advance the field significantly if it can be shown that SK11/12 and TTG1 work solely in the seed coat as a molecular switch to influence carbon partitioning in the seed.

>Authors:

We thank this reviewer for the encouraging comments. We have provided evidence and relevant discussions in the revised manuscript to show that SK11/12-mediated TTG1 phosphorylation maternally regulates carbon partitioning in seeds.

Reviewer 3

This paper builds an interesting gene regulatory model concerning regulation of major biosynthetic pathways in seeds by transcription factors. It would be of broad interest and use to plant scientists. The model is mostly well supported by the novel data presented. The experiments appear to be thorough and well designed.

>Reviewer:

Major comments

-I am convinced by the data that the proposed role of SK11 in the overall model (Fig 8) is justified. However, the role of SK12 is not as clear because of the differing band patterns within the phostag gels which indicate that SK12 phosphorylates different residues than SK11 during in vitro incubation with TTG1 in kinase assays. This is evident in fig 4a and 4d. Also the loading in 4d does not equal for the S215A lane which may have made it seem like there is a difference there.

>Authors:

We agree with this reviewer that SK11 and SK12 may not play exactly the same role in mediating TTG1. Based on the results of in vitro kinase assay shown in Fig. 4a,c,d, TTG1 phosphorylation levels mediated by SK12 is much lower than those by SK11. Further analyses of the potential phosphorylation sites by both LC-MS/MS (new Supplementary Fig. 11a,b) and in vitro kinase assay (Fig. 4c,d) have revealed that SK11 and SK12 target to multiple different sites, although Ser 215 is a common major target for both SK11 and SK12. These factors could contribute to different phosphorylation patterns of TTG1 mediated by SK11 and SK12 revealed by in vitro kinase assays. However, the major defects in seed characters were only found in *sk11 sk12* double mutants rather than in their single mutants. In addition, SK11 and SK12 at least target to the same Ser215 for phosphorylating TTG1. Overall, these results support that SK11 and SK12 play partially same roles in mediating phosphorylation of TTG1 to regulate carbon flow in *Arabidopsis* seeds.

As shown in Fig. 4a, TTG1 phosphorylation levels mediated by SK12 is rather low. To compare the relative levels of phosphorylated MBP-TTG1 to unphosphorylated MBP-TTG1 affected by various mutated sites in Fig. 4d, we have tried to adjust loadings based on comparable intensities of unphosphorylated MBP-TTG1 indicated by the lowest arrow on the Phos-tag gel. The loadings on the non-Phos-tag gel represented the actual amounts of MBP-TTG1 variants loaded on the Phos-tag gel shown above. This pattern will allow audience to easily compare the relative phosphorylated levels of MBP-TTG1 to comparable unphosphorylated MBP-TTG1 levels for various mutated sites.

>Reviewer

The phostag immunoblot in Fig 4b lacks controls which would enable an important comparison with 4a. Is the band pattern obtained from in vivo phosphorylation similar to that obtained during in vitro kinase assays with SK11 or SK12? The blot should contain samples similar to lanes 1, 3 and 4 in 4a.

>Authors:

Fig. 4a and 4b present mutually supportive results from two differently designed experiments. In vitro kinase assay shown in Fig. 4a involved various designed components required for in vitro phosphorylation in a cell-free system, for which we have included all necessary negative controls to exclude the side effects of individual components, such as MBP and ATP shown in lane 1. However, Fig. 4b shows the in vivo phosphorylation status of TTG1 using the protein extract from *ttg1-13 g4HA-TTG1* siliques. We have included two essential negative controls (not treated, NT; mock-treated, Mock) to clearly show the effect of bikinin treatment on the endogenous role of SK11 and SK12, which were present in plant protein extracts, in mediating TTG1 phosphorylation. Thus, all necessary negative controls have been included in two different experiments shown in Fig. 4a and 4b.

Unlike in vitro kinase assay in Fig. 4a using known and defined components, in vivo experiments in Fig. 4b may engage other unknown components relevant to the endogenous phosphorylation status of TTG1. For example, in addition to SK11 and SK12, TTG1 phosphorylation may also be affected by other kinases, which could result in an overall different phosphorylation pattern of TTG1 (Fig. 4b) from its pattern revealed in in vitro kinase assay (Fig. 4a). However, both results in Fig. 4a,b support that TTG1 is a substrate of GSK3-like kinases.

> Reviewer

-Fig 2b should include data from *ttg1* samples and ideally *tt2*?

>Authors:

As suggested by this reviewer, we have measured the expression of *SK11* and *SK12* in various tissues of *ttg1-13* and *tt2-5* as shown in new Supplementary Fig. 6. The general expression patterns of *SK11* and *SK12* in *ttg1-13* and *tt2-5* were similar to those in the wild-type background shown in Fig. 2a,b.

>Reviewer:

Minor comments

Introduction: -“...specifically mediates TTG1 effect on controlling carbon flow”. Other possible effects of this phosphorylation even are possible and not tested. Omit specifically.

>Authors:

We agree with this reviewer, and have deleted “specifically”.

>Reviewer:

-The term “master switch” seems unhelpful. What is a master switch vs a switch? Replace with control mechanism or something more precise.

>Authors:

As suggested by the reviewer, we have changed “master switch” to “important mechanism”.

>Reviewer:

Results:

-“...redundant biological functions” there is no evidence for this. Change to overlapping.

>Authors:

We have revised it as suggested.

>Reviewer:

-Can the developmental pattern of SK11 and TTG1 gene expression be interpreted with regards to the model? Is there a developmental change in carbon distribution between fatty acids and maternal tissue?

>Authors:

In developing wild-type seeds, expression of *SK11* and *SK12* remains at high levels at the embryo morphogenesis phase (0-4 days after pollination), but progressively decreases at the seed maturation phase (Fig. 2c,d). *TTG1* expression peaks in seeds 4 days after pollination, but decreases afterwards (Supplementary Fig. 6e). As revealed in this study, TTG1 and SK11/SK12 have antagonistic effects on seed characters, including fatty acid synthesis in the embryo, and production of mucilage and flavonoid pigments in the seed coat. Development of these seed characters is mostly initiated at the seed maturation phase (after 4 days after pollination), during which the embryo expands to fill the seed, whereas the endosperm is degraded to one cell layer surrounding the embryo (Baud et al., 2008; Francoz et al., 2015). Thus, site-specific phosphorylation of TTG1 by SK11/SK12 could allow these two antagonistic factors, which are all expressed at highest levels before proceeding to the seed maturation phase, to interact so as to balance subsequent carbon flow required for developing various seed characters at the seed maturation phase. In addition, we reason that distribution of carbon flow in the embryo and the seed coat should be developmentally controlled in the course of seed development to gradually develop mature seeds with cumulative storage reserves and protective seed coat.

Relevant references:

Baud et al. (2008) Storage reserve accumulation in Arabidopsis: metabolic and developmental control of seed filling. The Arabidopsis Book 6, e0113.

Francoz et al. (2015) Arabidopsis seed mucilage secretory cells: regulation and dynamics. Trends Plant Sci. 20, 515-524.

>Reviewer:

-Can the statistical analysis of figure 2b and g be done with ANOVO and post-hoc testing in order to compare all samples (e.g. indicated with different letters)

>Authors:

As suggested by this reviewer, we have performed the requested statistical analyses and provided these data in relevant panels in Fig. 2 and new Supplementary Fig. 6.

>Reviewer:

-“...compromised TTG1 binding to the GL2 locus” Change compromise to weakened.

>Authors:

We have revised it as suggested.

>Reviewer:

“This expression trend generally followed that of TTG1 and TT2” The patterns between GL2 and TTG1 is not similar.

>Authors:

We agree with the reviewer, and have revised the sentence to more precisely describe the *GL2* expression.

>Reviewer:

-Stats should be applied to Fig 7g.

>Authors:

As suggested by this reviewer, we have performed statistical analysis and provided the data in new Fig. 7g.

>Reviewer:

Discussion:

-The mechanism of TTG1 is referred to several times as carbon partitioning between metabolic pathways. Is it not more likely to represent carbon partitioning between tissues (maternal vs embryo)? For example what about seeds which don't store much oil?

>Authors:

We have performed reciprocal genetic crosses between WT and *sk11 sk12*, and found that SK11 and SK12 maternally regulate fatty acid levels (new Supplementary Fig. 17). In addition, previous studies have also shown the maternal effects of TTG1 and GL2 on fatty acid levels in seeds (Chen et al., 2015; Shi et al., 2012), and that TT2 is predominantly expressed in the seed coat and controls seed coat development (Nesi et al., 2001; Gonzalez et al., 2009). Thus, all these findings support that SK11/12-mediated TTG1 phosphorylation and its interaction with TT2 as well as their effects on *GL2* maternally regulate fatty acid levels in seeds. Since fatty acid synthesis in the embryo, and production of mucilage and flavonoid pigments in the seed coat share the same carbon source, our results suggest that site-specific phosphorylation of TTG1 by SK11/SK12 maternally modulates carbon partitioning among metabolic pathways in zygotic and maternal tissues in seeds. We have incorporated the new results and relevant discussions in the revised manuscript. At the stage, we do not have much information to reason whether our model in *Arabidopsis* is applicable to other seeds without much oil.

Relevant references:

Chen et al. (2015) TRANSPARENT TESTA GLABRA 1 regulates the accumulation of seed storage reserves in *Arabidopsis*. *Plant Physiol.* 169, 391-402.

Shi et al. (2012). *Arabidopsis glabra2* mutant seeds deficient in mucilage biosynthesis produce more oil. *Plant J.* 69, 37-46.

Nesi et al. (2001) The *Arabidopsis* TT2 gene encodes an R2R3 MYB domain protein that acts as a key determinant for proanthocyanidin accumulation in developing seed. *Plant Cell* 13, 2099-2114.

Gonzalez et al. (2009) TTG1 complex MYBs, MYB5 and TT2, control outer seed coat differentiation. *Dev. Biol.* 325, 412-421.

>Reviewer:

-Is Ser215 of TTG1 conserved among plants? The adjacent proline may also be involved in directing phosphorylation.

>Authors:

As suggested by this reviewer, we have analyzed the protein sequences of TTG1 homologs from various plant species, and found that Ser215 and the adjacent Pro216 are well conserved in these proteins (new Supplementary Fig. 14). Whether this Pro216 is also involved in mediating phosphorylation of TTG1 needs to be further investigated.

Reviewers' comments:

Reviewer #1 (Remarks to the Author):

Li and colleagues have added an impressive amount of new data to this manuscript, and have vastly improved the quality of the work presented. The majority of my main queries have been addressed to my entire satisfaction. In particular I now find the genetic analysis globally clear and well argued, and I feel that the story being told by the authors is much more clearly presented and that their conclusions are much better supported than in the previous version.

I do, however, have some points which I feel need to be addressed

1) Figure 1a and Supplementary Fig. 18.

In these pull-down assays the "input" lanes appear to have been run on a different gel to the pull-down assays (from the Ponceau staining in Supplementary Fig. 18). I don't fully understand what is going on in this figure but this needs to be clarified.

2) Page 8, fourth line from the top

Fig 2b does not show expression patterns, but expression levels in different tissues. Please change "The expression patterns of....." to "The global expression levels of....."

3) Page 8 and Supplementary Figure 7

It seems pretty clear in Supplementary Figure 7 that SK11 and possibly also SK12 expression is excluded from the outer layer of the seed coat at the stage shown. This is a bit odd given that these genes have an affect on mucilage production. The observations are certainly not irreconcilable but possibly a sentence or two addressing this would be helpful.

4) Supplementary Figure 17. IMPORTANT!

Unless I am very much mistaken, this figure, and the results presented in it (reciprocal crosses) are not presented in the results section but only in the discussion. This needs to be addressed. Furthermore, the Fatty acid contents of the sk11/sk12 double mutant in this figure are very different to those in Figure 3b, and not as strikingly different to those in wild-type as in Figure 3b. For the statistics sample sizes are not provided, which is unsatisfactory, and globally the results, although apparently statistically significant are not very convincing. I wonder whether the inverse experiment, of pollinating the SK11:SK11E292KGFP line with wild-type pollen, and checking seed coat phenotypes, would not provide a useful confirmation of the results presented. The results of these crosses are important to the impact of this paper (thus meriting inclusion in the results section!) and I think this point, which was pointed out by all three reviewers, is very important.

Reviewer #2 (Remarks to the Author):

I have reviewed the authors' responses to my comments and the comments of the other reviewers. I believe that the authors have addressed all of the concerns adequately.

Reviewer #3 (Remarks to the Author):

The revised manuscript is greatly improved. The authors have adequately dealt with my previous comments. The major claims of the paper have overall been strengthened, which should have the effect of increasing the influence of the paper. I continue to believe the paper would be of wide interest to plant scientists.

The one anomaly that I perceive to remain in the results is the issue with Figure 6b raised by another reviewer surrounding the statement: "In vitro pull-down assays demonstrated that TT2-His had a stronger interaction with MBP-TTG1S215A than MBP-TTG1, but did not interact with MBP-TTG1S215E (Fig. 6b). "

The recombinant MBP-TTG1 is not at all phosphorylated according to Figure 4a. Nor would you expect a recombinant protein expressed from bacteria to be phosphorylated as in planta. So how come the interaction between MBP-TTG1 and TT2 is less than with the MBP-TTG1 Ser215Ala mutant if neither are phosphorylated? Such a result would indicate that changes in interaction strength in this case are not due to phosphorylation/dephosphorylation. This result weakens (but does not dismiss) support for the overall model and should be discussed.

Response to Reviewers

We would like to thank the three reviewers for the time committed in reviewing the revised manuscript, and for all the detailed suggestions on improving this manuscript. We have further revised the manuscript to fully address the reviewers' concerns and criticisms as follows.

Reviewer 1

Li and colleagues have added an impressive amount of new data to this manuscript, and have vastly improved the quality of the work presented. The majority of my main queries have been addressed to my entire satisfaction. In particular I now find the genetic analysis globally clear and well argued, and I feel that the story being told by the authors is much more clearly presented and that their conclusions are much better supported than in the previous version.

I do, however, have some points which I feel need to be addressed.

>Reviewer:

Figure 1a and Supplementary Fig. 18.

In these pull-down assays the “input” lanes appear to have been run on a different gel to the pull-down assays (from the Ponceau staining in Supplementary Fig. 18). I don't fully understand what is going on in this figure but this needs to be clarified.

>Authors:

As the amount of the input (prey proteins) is much lower than that of bait proteins (GST and GST-SK11/12) in our GST pull-down assay, we only performed Ponceau S staining for showing the loading amount of the bait proteins in original Fig. 1a. As requested by this reviewer, we have repeated this GST pull-down experiment and included the result of Ponceau S staining of both input prey proteins and bait proteins in the new Supplementary Fig. 17. Consistently, Fig. 1a is now replaced accordingly in the revised manuscript.

>Reviewer:

Page 8, fourth line from the top

Fig 2b does not show expression patterns, but expression levels in different tissues. Please change “The expression patterns of.....” to “The global expression levels of.....”

>Authors:

We thank the reviewer for this correction, and have revised the relevant description accordingly.

>Reviewer:

Page 8 and Supplementary Figure 7

It seems pretty clear in Supplementary Figure 7 that SK11 and possibly also SK12 expression is excluded from the outer layer of the seed coat at the stage shown. This is a bit odd given that these genes have an effect on mucilage production. The observations are certainly not irreconcilable but possibly a sentence or two addressing this would be helpful.

>Authors:

Both *SK11* and *SK12* are expressed in the entire seed coat (integument) based on the GUS staining patterns shown in Fig. 2e and the data from the Arabidopsis eFP browser (Supplementary Fig. 7b). This expression pattern is consistent with their role in affecting mucilage production.

>Reviewer:

Supplementary Figure 17. IMPORTANT!

Unless I am very much mistaken, this figure, and the results presented in it (reciprocal crosses) are not presented in the results section but only in the discussion. This needs to be addressed. Furthermore, the Fatty acid contents of the *sk11/sk12* double mutant in this figure are very different to those in Figure 3b, and not as strikingly different to those in wild-type as in Figure 3b. For the statistics sample sizes are not provided, which is unsatisfactory, and globally the results, although apparently statistically significant are not very convincing. I wonder whether the inverse experiment, of pollinating the *SK11:SK11E292K:GFP* line with wild-type pollen, and checking seed coat phenotypes, would not provide a useful confirmation of the results presented. The results of these crosses are important to the impact of this paper (thus meriting inclusion in the results section!) and I think this point, which was pointed out by all three reviewers, is very important.

>Authors:

As suggested by this reviewer, we have included the original Supplementary Fig. 17 as part of new Fig. 3g (middle panel), and also provided the results of reciprocal crosses between *SK11:SK11^{E292K}-GFP* and wild-type plants in the right panel of Fig. 3g. The results of reciprocal genetic crosses between *sk11 sk12* and wild-type plants and between *SK11:SK11^{E292K}-GFP* and wild-type plants both support a maternal effect of SK11 and SK12 on regulating fatty acid levels in seeds. Measurement of fatty acid levels were all preformed on three biological replicates, which is described in Fig. 3g legend.

The reviewer also asked why there was a variation in fatty acid levels of *sk11 sk12* seeds in original Fig. 3b and Supplementary Fig. 17. As these seeds were collected in different batches and measured for their fatty acid levels in independent processes, a variation in their absolute values is normal. The most important result is that the fatty acid levels of *sk11 sk12* seeds are consistently lower than those of wild-type seeds in both original Fig. 3b and Supplementary Fig. 17 (now the middle panel of Fig. 3g).

Reviewer 2

I have reviewed the authors' responses to my comments and the comments of the other reviewers. I believe that the authors have addressed all of the concerns adequately.

Reviewer 3

The revised manuscript is greatly improved. The authors have adequately dealt with my previous comments. The major claims of the paper have overall been strengthened, which should have the effect of increasing the influence of the paper. I continue to

believe the paper would be of wide interest to plant scientists.

>Reviewer:

The one anomaly that I perceive to remain in the results is the issue with Figure 6b raised by another reviewer surrounding the statement: "In vitro pull-down assays demonstrated that TT2-His had a stronger interaction with MBP-TTG1S215A than MBP-TTG1, but did not interact with MBP-TTG1S215E (Fig. 6b). "

The recombinant MBP-TTG1 is not at all phosphorylated according to Figure 4a. Nor would you expect a recombinant protein expressed from bacteria to be phosphorylated as in planta. So how come the interaction between MBP-TTG1 and TT2 is less than with the MBP-TTG1 Ser215Ala mutant if neither are phosphorylated? Such a result would indicate that changes in interaction strength in this case are not due to phosphorylation/dephosphorylation. This result weakens (but does not dismiss) support for the overall model and should be discussed.

>Authors:

Our in vitro pull-down assays (Fig. 6b) demonstrated that TT2-His had a stronger interaction with MBP-TTG1^{S215A} than MBP-TTG1. As MBP-TTG1 is not phosphorylated in our bacterial expression system (Fig. 4a), we agree with this reviewer that this difference in the interaction between TT2-His and MBP-TTG1^{S215A} or MBP-TTG1 could be irrelevant to the effect of phosphorylation of TTG1 at Ser 215. Instead, mutation of S to A at Ser 215 in TTG1 may also affect other biochemical characteristics that influence the interaction between TTG1 and TT2. As suggested by this reviewer, we have incorporated the relevant discussion along with the description of Fig. 6b results. Nevertheless, this observation does not affect the main conclusion that phosphorylation of TTG1 at Ser 215 abolishes the interaction between TTG1 and TT2, which is supported by all results describing the effects of the phosphorylation-mimicking mutation of S to E at Ser 215 (Fig. 6a-d).

Reviewers' comments:

Reviewer #1 (Remarks to the Author):

I am satisfied with the authors comments to most of my queries. However I do not agree that the results in Figure 2E support expression in the outer cell layer of the outer integument (where mucilage secreting cells differentiate). There is clearly NO GUS labelling in this outer cell layer for SK11, and very reduced, if any labelling for SK12. The argument that the eFP browser shows expression throughout the integuments is completely spurious, since this data is generated from a pooled "seed coat" sample that gives no spatial resolution.

This point requires further discussion.

Reviewer #3 (Remarks to the Author):

The authors have adequately addressed my comments.

Response to Reviewers

We would like to thank the three reviewers for the time committed in reviewing the revised manuscript in the last two rounds. We have further revised the manuscript to address Reviewer 1's question as follows.

Reviewer 1

I am satisfied with the authors' comments to most of my queries. However I do not agree that the results in Figure 2E support expression in the outer cell layer of the outer integument (where mucilage secreting cells differentiate). There is clearly NO GUS labelling in this outer cell layer for SK11, and very reduced, if any labelling for SK12. The argument that the eFP browser shows expression throughout the integuments is completely spurious, since this data is generated from a pooled "seed coat" sample that gives no spatial resolution. This point requires further discussion.

>Authors:

We agree with this reviewer that eFP browser data are not completely accurate, and would like to provide the following explanation and the relevant new figure panels to address the question why *SK11* is not expressed in the outer layer of the seed coat shown in original Fig. 2e as below.

The outmost layer of mature Arabidopsis seeds consists of dead cells and mucilage constituents. At an early stage of seed development (e.g. 2 days after pollination, globular stage under our conditions), *SK11/12* are indeed expressed in the outermost layer of the seed coat. We have provided the representative GUS staining images in the revised Fig. 2e (left panels) and other similar results in the revised Supplementary Fig. 7b. At the heart stage (4 days after pollination under our conditions), the outermost cell layer of the seed coat is being highly specialized with gradual accumulation of mucilage constituents and starch granules while losing other normal cellular functions (Western et al., 2000; Francoz et al., 2015). This may explain why expression of *SK11/12* is mostly undetectable in the outermost cell layer of the seed coat at the heart stage shown in the previous Fig. 2e.

Relevant References:

Western et al. (2000) Differentiation of mucilage secretory cells of the Arabidopsis seed coat. Plant Physiol 122, 345-356.

Francoz et al. (2015) Arabidopsis seed mucilage secretory cells: regulation and dynamics. Trends in Plant Sci 20, 515-524.

Reviewer 2

I have reviewed the authors' responses to my comments and the comments of the other reviewers. I believe that the authors have addressed all of the concerns adequately.

Reviewer 3

The authors have adequately addressed my comments.

Manuscript Number: NCOMMS-16-30680C

Response to Reviewers' Comments

All reviewers' comments have been addressed in previous versions of the manuscript.